# VIDEO-STAR: SELF-TRAINING ENABLES VIDEO INSTRUCTION TUNING WITH ANY SUPERVISION

**Orr Zohar**[1†]**, Xiaohan Wang**[1]**, Yonatan Bitton**[2]**, Idan Szpektor**[2] **& Serena Yeung-Levy**[1†]
[1]Stanford University, [2]Google Research
[†]{orrzohar,syyeung}@stanford.edu
Project page: https://orrzohar.github.io/projects/video-star/

## ABSTRACT

The performance and reasoning capabilities of Large Multi-modal Models (LMMs) is dependent on the size and quality of their training datasets. However, collecting datasets that support chain-of-thought instruction tuning is highly challenging. Existing video instruction tuning datasets are often derived by prompting large language models with video captions to generate question-answer pairs, which makes them predominantly descriptive rather than reasoning-focused. Meanwhile, many labeled video datasets with diverse labels and supervision exist – however, we find that their integration into LMMs is non-trivial. To address this, we introduce Video Self-Training with augmented Reasoning (Video-STaR), the first self-training approach for video instruction tuning. Video-STaR allows the utilization of *any* labeled video dataset for video instruction tuning. In Video-STaR, an LMM cycles between instruction generation and finetuning, which we show (I) improves general video understanding and (II) adapts LMMs to novel downstream tasks with existing supervision. During instruction generation, an LMM is prompted to propose an answer. The answers are then filtered only to those that contain the original video labels, and the LMM is then re-trained on the generated dataset. By training exclusively on generated answers containing the correct video labels, Video-STaR leverages these existing labels as weak supervision for video instruction tuning. Our results demonstrate that Video-STaR-augmented LMMs achieve notable improvements in (I) general Video QA, where TempCompass performance improved by 6.1%, *and* (II) downstream tasks, with a 9.9% increase in Kinetics700-QA accuracy and a 4.0% improvement in action quality assessment on FineDiving, while also exhibiting better interpretability.

## 1 INTRODUCTION

The advent of Large Multi-modal Models (LMMs) marked a significant milestone in artificial intelligence. These models aim to create versatile systems capable of understanding and executing vision-and-language tasks aligned with human intentions. Liu et al. (2023a) demonstrated the importance of visual instruction tuning on the resulting LMM's performance. While significant advancements have been made with image-based LMMs, video-LMMs continue to face challenges due to the increased complexity of videos, which involve more intricate scene dynamics and temporal information. This complexity demands larger, more diverse, and reasoning-focused video instruction tuning datasets. However, the largest existing video instruction dataset, VideoInstruct-$100K$ (VI-$100K$) (Maaz et al., 2023), comprises $100K$ video-text pairs but only $13K$ unique videos. This is small compared to image instruction datasets like Cambrian-10M (Tong et al., 2024), which contain millions of image question pairs from diverse domains and tasks.

Furthermore, due to video instruction tuning dataset construction - mainly prompting large language models to produce question-answer pairs - these video datasets often degrade to simplistic questions, prompting for video captions — 75% of VI-$100K$'s questions are of this type (see App. Fig. 10), lacking diversity and reasoning. Combining different sources of supervision has the potential to generate more diverse video instruction tuning datasets, enhancing video understanding. Such supervision exists, as the broader computer vision community has developed an extensive collection of video benchmarks tailored for diverse tasks such as action recognition (Smaira et al., 2020; Soomro

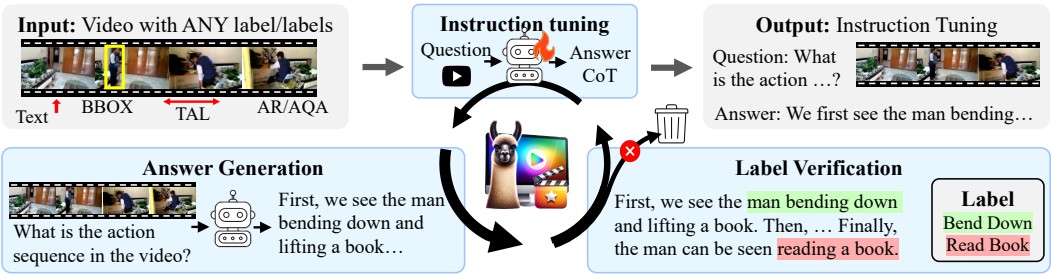

Figure 1: **Video-STaR Overview.** Video-STaR can utilize *any* labeled video dataset, including AR (Action Recognition), AQA (Action Quality Assessment), and TAL (Temporal Action Localization) – from which it generates video instruction tuning data (video, question, answer triplets). Internally, Video-STaR cycles between: (I) **Answer Generation**, where an LMM is prompted to generate candidate answers for the questions. (II) **Label Verification** where generated answers are filtered to only those that contain the video labels. And (III) **Instruction Tuning**, where a model is retrained on answers that pass verification. These cycles continue until performance plateaus, producing both an instruction tuning dataset and an improved LMM.

et al., 2012), action quality assessment (Xu et al., 2022b; Zhang et al., 2023c), among others (Wu et al., 2021; Grunde-McLaughlin et al., 2021; Grauman et al., 2023; Luo et al., 2022).

Beyond improving overall LMM performance, adapting LMMs to novel or out-of-domain tasks is also crucial. While LMMs have many novel and impactful applications, many remain out of reach — such as analyzing radiology images (Senkaiahliyan et al., 2023), meteorological data (Lawson et al., 2024), traffic analysis (Zhou & Knoll, 2024), judging sporting events (e.g., gymnastics, Olympic diving), and assisting in surgical procedures, among others (Zheng et al., 2024; Jiang et al., 2024; Deng et al., 2024). These tasks require expert, in-domain knowledge that LMMs lack, necessitating adaptation through instruction tuning. However, collecting video instruction tuning datasets is complex and requires extensive manual effort. For instance, training an 'AI judge' to judge Olympic diving would traditionally involve collecting detailed expert critiques of each dive. On the other hand, these tasks often include auxiliary annotations that could be leveraged, such as surgical outcomes in medical procedures or judging scores in Olympic events.

To address these challenges, we take inspiration from LLM's capability for self-improvement (a.k.a self-training) (Huang et al., 2022; Zelikman et al., 2022; Singh et al., 2023), which involves training a model on its generated data and filtering to exclude low-quality outputs. A model's performance is improved by cycling between generation, filtering, and training. Inspired by the success of self-training in LLMs, we hope that applying self-training to LMMs could improve their performance on complex reasoning and multi-modal tasks. Herein, we explore self-training in LMMs and introduce Video Self-Training with augmented Reasoning (Video-STaR, see Fig. 1). Video-STaR enables the incorporation of *any* labeled video dataset of any format or task by prompting an LMM with the video and a question to generate answers (Fig. 2, 2.1) containing the video content's label, allowing the utilization of these labels as form of weak supervision. If the model cannot correctly answer the question, we provide the original video label and ask it to rationalize it (Fig. 2, 2.2). We then again reject answers that do not contain the gold label (Fig. 2, 2.3). By facilitating the use of any supervision for video instruction tuning, Video-STaR enables the creation of diverse datasets with chain-of-thought reasoning.

Our experimental setup initializes Video-STaR with Video-LLaVA (Lin et al., 2023), focusing on assessing its impact on video question-answering (VQA) performance. After a few Video-STaR training cycles, we compare the performance of Video-STaR to other LMMs and strong baselines, which utilize the entire source datasets, to gauge the effectiveness of the Video-STaR framework. Our findings demonstrate notable enhancements in accuracy and reasoning capabilities, highlighting Video-STaR's role in overcoming the constraints posed by conventional video instruction tuning datasets. We show that the integration of Video-STaR not only boosts Video-LLaVA's performance on standard zero-shot VQA benchmarks but also significantly improves its adaptability to various downstream video understanding tasks. This underscores Video-STaR's capacity to advance LMM training while improving overall performance and versatility.

**Our contributions can be summarized as follows:**

1. We introduce Video Self-Training with augmented Reasoning (Video-STaR), the first video self-training method for Large Multi-modal Models. Video-STaR enables the use of *any* labeled video dataset for visual instruction tuning with chain-of-thought reasoning.

2. Video-STaR improves zero-shot video question answering performance on various benchmark datasets, compared to strong baselines, as evidenced by increased accuracy on TempCompass from $45.7\%$ to $51.8\%$.

3. We demonstrate that Video-STaR can adapt LMMs to diverse video tasks, notably enhancing action quality assessment accuracy on FineDiving from $17.6\%$ to $21.6\%$. Additionally, with Video-STaR, we show that LMMs can also explain *why* a score is given.

4. Utilizing Video-STaR, we create a large, $1M$ video instruction tuning dataset with rich chain-of-thought reasoning - VSTAR-1M, sourced from diverse datasets and tasks, and show that it benefits LMM performance.

## 2    VIDEO SELF-TRAINING WITH AUGMENTED REASONING (VIDEO-STAR)

Given a dataset of videos $v$ and their corresponding labels $l : \mathcal{D} = \{(v_i, l_i)\}_{i=1}^{d}$, Video-STaR's objective is to create question $q$ and chain-of-thought answer $a$, that contains both the rationale $r$ and final answer ($a = r \cup l$). These pairs are then used to instruction-tune the pre-trained model $M$ on the dataset $\hat{\mathcal{D}} = \{(v_i, q_i, a_i)\}_{i=1}^{d_f}$, producing the instruction-tuned model $\hat{M}$. Note that videos need not be from the same task, and may contain multiple labels. We start by prompting a large language model with a task description $T$ and video labels $L$ to generate candidate questions $q$:

$Y_{T,L} =$ A video is labeled {L} for the task of {T}. What questions could you ask someone about the video that should contain the video labels in the response?

Video-STaR performs generation-training cycles, where in cycle $i$ the instruction-tuned model $\hat{M}^{i\star}$ is produced, while the instruction-tuned model from the previous cycle $\hat{M}^{(i-1)\star}$ is utilized for training data generation. We initialize the process with $\hat{M}^{0\star}$, an existing instruction-tuned model.

To prepare the training data in cycle $i$, answers are generated either directly via *Answer Generation* or through backward rationalization via *Label Rationalization*. In *Answer Generation*, $\hat{M}^{(i-1)\star}$ is prompted with questions (Sec. 2.1). Candidate answers are then filtered using the original video labels (Sec. 2.3). Videos rejected during direct Answer Generation are rationalized, where $\hat{M}^{(i-1)\star}$ is provided both a video $v_i$ and labels $l_i$, and then prompted with the question again (Sec. 2.2). Candidate answers are filtered again, creating the instruction tuning dataset in cycle $i$, $\hat{\mathcal{D}}_i$ of size $d_i$. A pre-trained model $M$ is then finetuned on $\mathcal{D}_i$, producing $\hat{M}^{i\star}$. The next cycle generates data using $\hat{M}^{i\star}$, until the performance plateaus (see Fig. 2).

### 2.1    ANSWER GENERATION

Each Video-STaR cycle begins in direct Answer Generation. In this phase, $\hat{M}^{(i-1)\star}$ is prompted with the video-question pair to provide an answer along with a detailed rationale:

$Y_Q =$ Question: {Q}. Rationalize your answer step-by-step; how can one arrive at this conclusion?

When prompted with the question $q_i$ on a particular video, $\hat{M}^{(i-1)\star}$ is expected to generate an answer $a_i$ that contains the label $\hat{l}_i$ and the rationale $r_i$ ($a_i = r_i \cup \hat{l}_i$, see Fig. 2). As can be seen in App Fig. 13, 12, answers containing the correct labels are of higher quality and suffer less from hallucination. Therefore, we filter the generated answers to include only those that contain the correct label utilizing the verifier (Sec. 2.3). For an example of Answer Generation, see Fig. 3.

### 2.2    LABEL RATIONALIZATION

Answer generation has two main drawbacks: (i) In some applications, especially on challenging/out-of-domain tasks, initial Answer Generation yield is low, resulting in almost no training samples after filtering (e.g., FineDiving, see Fig. 3); (ii) improvement plateaus as the model fails to solve new problems in the training set, and it is only trained on examples it answers correctly.

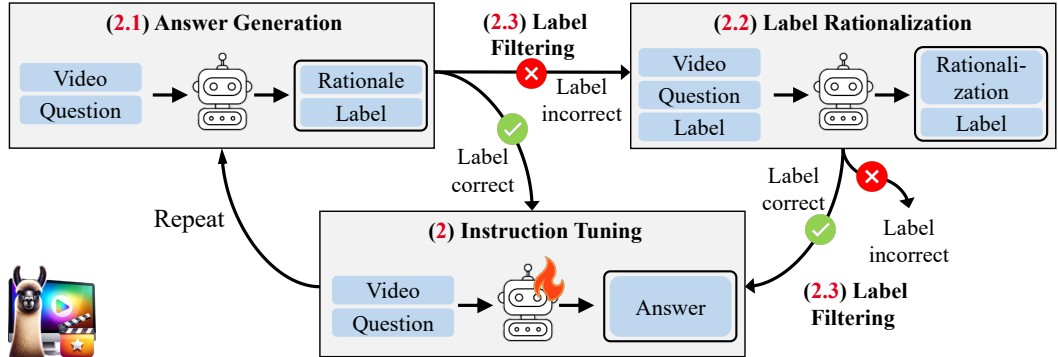

Figure 2: **Video Self-Training with augmented Reasoning. (2.1)** We initialize by prompting an LMM to generate an answer for a particular video. **(2.3)** We then filter the generated answers to those only containing the original video labels. **(2.2)** The videos whose generated answer did not contain the ground-truth labels are then sent to label rationalization, where given the video, question, and label - the model is expected to rationalize the label. **(2.3)** The generated answers are filtered again to those only containing the ground-truth labels, and **(2)** the LMM is instruction-tuned from the pre-trained checkpoint on the resulting dataset. The cycle is then repeated.

Inspired by Zelikman et al. (2022), for videos whose $\hat{M}^{(i-1)\star}$ generated answers did not contain the ground-truth labels, we introduce *label rationalization* as part of Video-STaR. Concretely, we provide $\hat{M}^{(i-1)\star}$ the video, question, and video label and instruct the model to rationalize the label:

$Y_{Q,L}$ = Question: {Q}. Answer: {L}.
Can you rationalize the answer step-by-step? How can one arrive at this conclusion?

Given the correct label, the model can reason backward and more easily generate a rationale leading to the correct answer. However, Label Rationalization is more prone to hallucinations, so we prefer direct Answer Generation and use rationalizations only if answer generation fails (see App. Sec. F). For an example of Label Rationalization, see Fig. 3. The generated answers are then filtered, keeping only those that contain the gold label ($\hat{l}_i = l_i$) utilizing the verifier (Sec. 2.3). Label Rationalization is only utilized in training cycles; only Answer Generations produce the final model $\hat{M}^\star$.

## 2.3 LABEL VERIFICATION

Video-STaR aims to utilize the labels as weak supervision in instruction tuning data generation. Gold labels are a grounding aspect of our datasets and represent some ground-truth knowledge. In App. Sec. F.1, we show that answers that contain the ground-truth labels in their responses are of higher quality than those that don't and have a higher probability of being correct. While we would like to validate the existence of the different labels in the generated text, this can be non-trivial.

To this end, we introduce the Parser-Verifier. The *Parser*, $P$ extracts the predicted labels from the generated text ($\hat{l}_i = P(a_i)$), using a mixture of named entity recognition and Regex. Regex is used to identify easily identifiable string patterns, such as bounding boxes and time ranges, while named entity recognition is used for more nuanced entities, such as timestamps. The *Verfier*, $V$ compares the extracted labels with the gold ones using the appropriate metrics ($V(l_i, \hat{l}_i) \to \mathbb{R}$). For example, IoU for bounding boxes/temporal action localization, and BERT (Devlin et al., 2018) embedding similarity for sentence ordering. Each video has between 1-3 associated labels. To be classified as correct, the predicted labels must be within a 5% margin of error from the gold. See App. Tab. 8 for a comprehensive list of the label types, their description, and corresponding parsers and verifiers.

## 3 VIDEO-STAR GENERATED DATASET - VSTAR-1M

In this section, we detail the different source datasets utilized in our study (Sec. 3.1) and analyze the generated Video-STaR Dataset, VSTAR-1M (Sec. 3.2).

| Source | Videos | Labels | Avg. Dur. | Source Task |
|---|---|---|---|---|
| Kinetics700 (2020) | 650K | 700 | 5.1 | Action Recognition |
| STAR benchmark (2021) | 22K | 207 | 28.6 | Video Reasoning, Temporal Action Localization, Bounding Box |
| FineDiving (2022b) | 3K | 1065 | 3.2 | Action Quality Assessment, Action Sequence |

Table 1: **Source Dataset Summary.** Video datasets used as source datasets for instruction tuning data generation, their source tasks, and the number of unique labels.

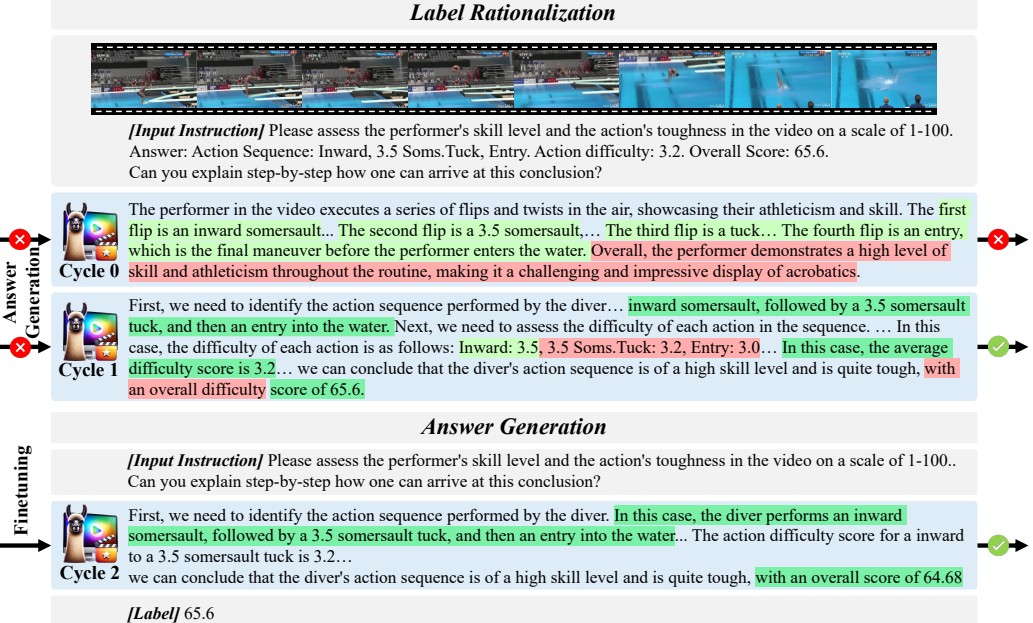

Figure 3: **Qualitative Improvement of Data Generation over Cycles on FineDiving.** We initialize the model with Video-LLaVA (Cycle 0), where the model cannot generate an answer ($\rightarrow |\times$) or rationalize the label correctly ($| \rightarrow \times$). In the second cycle (Cycle 1), the model still cannot generate an answer ($\rightarrow |\times$) but can rationalize the video label ($\checkmark | \rightarrow$), which is selected for instruction tuning. Finally, in the third cycle (Cycle 2), the model directly generates a correct answer ($\checkmark | \rightarrow$), which is selected for visual instruction tuning. We highlight in green correct answers, in red wrong answers, and in yellow - hallucinations.

## 3.1 SOURCE DATASETS

In selecting source datasets, we selected datasets that contain diverse video content and label types, please see Tab. 1. These include Kinetics700 (Smaira et al., 2020), which has action recognition annotations and is particularly large and diverse. FineDiving (Xu et al., 2022b) is an action quality assessment dataset of Olympic diving events and has both an overall score and action sequence annotations. Finally, STAR-benchmark (Wu et al., 2021), a video reasoning dataset, also contains bounding box and temporal action localization annotations. Tab. 1 contains the relevant dataset statistics, e.g., the number of videos and labels per dataset.

## 3.2 GENERATED DATASET ANALYSIS

**Quantitative Analysis.** Through the application of Video-STaR, significant dataset augmentation was achieved over two cycles of, illustrated in Fig. 4. This figure displays the Answer Generation

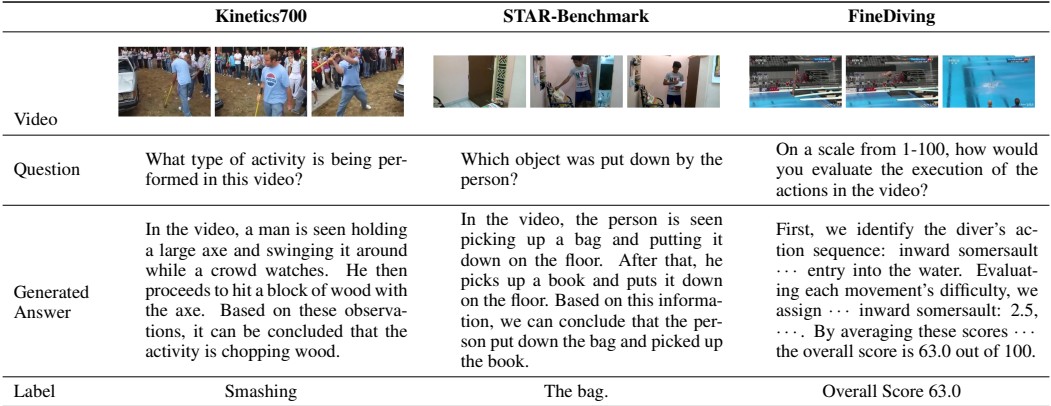

| | **Kinetics700** | **STAR-Benchmark** | **FineDiving** |
|---|---|---|---|
| Question | What type of activity is being performed in this video? | Which object was put down by the person? | On a scale from 1-100, how would you evaluate the execution of the actions in the video? |
| Generated Answer | In the video, a man is seen holding a large axe and swinging it around while a crowd watches. He then proceeds to hit a block of wood with the axe. Based on these observations, it can be concluded that the activity is chopping wood. | In the video, the person is seen picking up a bag and putting it down on the floor. After that, he picks up a book and puts it down on the floor. Based on this information, we can conclude that the person put down the bag and picked up the book. | First, we identify the diver's action sequence: inward somersault ··· entry into the water. Evaluating each movement's difficulty, we assign ··· inward somersault: 2.5, ····. By averaging these scores ··· the overall score is 63.0 out of 100. |
| Label | Smashing | The bag. | Overall Score 63.0 |

Table 2: **Examples of Generated Data.** Examples of the video, question, Video-STaR generated answer, and ground-truth label from each source dataset.

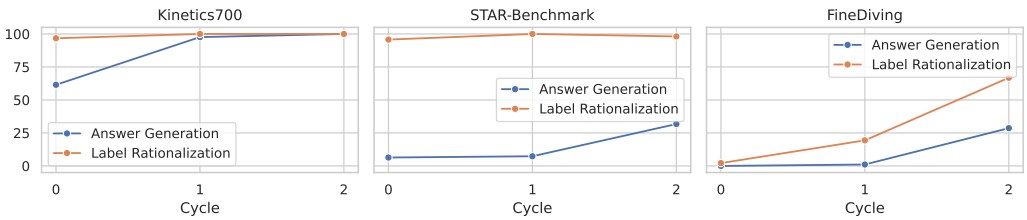

Figure 4: **Dataset Yield vs. Cycles.** Percentage of the videos converted to instruction tuning by the Answer Generation and Label Rationalization per dataset. As can be seen, on difficult datasets, such as FineDiving, no videos are converted by Answer Generation in the first cycle. By utilizing Label Rationalization, the model is able to improve to eventually generate answers correctly.

and Label Rationalization yield across the datasets source. Notably, the initial application of Video-LLaVA on datasets like Kinetics700 and STAR-Benchmark showed significant Answer Generation success rates. However, the FineDiving dataset presented a notable challenge, with Answer Generation having no answers generated directly, underscoring the complexity of the dataset and the critical role of Label Rationalization. By the end of the second cycle, a substantial number of high-quality instances had been produced, showcasing both the effectiveness of Video-STaR in converting labeled video datasets into video instruction tuning dataset, as evidenced in Fig. 4.

**Qualitative Analysis.** See Tab. 2 for examples of generated question-answer pairs. From Kinetics700, we extracted an instance showcasing a video labeled 'smashing'. Video-STaR correctly identified a more fine-grained label, 'chopping wood'. In the FineDiving dataset, a clip depicting a complex dive was accompanied by the question 'On a scale from 1-100···' The model's output text provided a breakdown of the dive's components, leading to a score (label), as would be desired from an LMM visual assistant. Finally, in the STAR benchmark, questions are already provided; therefore, we utilized them directly.

In 3, we show the qualitative improvement of the generated data over Video-STaR cycles. In the first cycle (Cycle 0), Video-LLaVA failed at Answer Generation and Label Rationalization. After one Video-STaR cycle (Cycle 1), Video-STaR still failed at Answer Generation but succeeded in Label Rationalization. After the final Video-STaR cycle (Cycle 2), Video-STaR managed to generate the answer without requiring the label via Answer Generation.

# 4 EXPERIMENTS

We experimented with Video-STaR and evaluated its enhanced video understanding capabilities. In Sec. 4.3, we evaluate how Video-STaR adapts Large Multi-modal Models (LMMs) to the source datasets and how these capabilities are transferred zero-shot to similar benchmarks. In Sec. 4.2, we evaluate the video question-answering capabilities on video benchmark datasets.

| | Action | | Direction | | Speed | | Event | Attribute Change | | | | Avg. |
|---|---|---|---|---|---|---|---|---|---|---|---|---|
| | Fine | Coarse | Obj. | Cam. | Abs. | Rel. | Order | Color | Size | Both | Other | |
| Random | 39.7 | 40.1 | 39.8 | 39.0 | 40.8 | 42.0 | 41.5 | 40.4 | 39.9 | 38.9 | 39.4 | 40.5 |
| mPLUG-Owl (2023) | 48.8 | 66.1 | 38.7 | 36.8 | 42.2 | 38.4 | 42.0 | 41.7 | 44.7 | 41.9 | 39.9 | 44.4 |
| Video-LLaVA (2023) | 63.4 | 93.5 | 36.1 | 34.8 | 42.7 | 26.5 | 39.1 | 52.6 | 37.1 | 43.3 | 33.3 | 45.7 |
| Video-LLaVA$^+$ | 62.1 | 93.0 | 35.0 | 32.6 | 41.1 | 38.7 | 36.4 | 59.0 | 40.2 | 36.7 | 44.4 | 47.2 |
| Vid-LLaVA$^{Gemini}$ | 65.7 | 90.5 | 38.9 | 46.0 | 41.8 | 42.4 | 41.0 | 48.7 | 49.1 | 32.5 | 40.4 | 48.7 |
| Video-STaR | 68.6 | 94.1 | 39.9 | 39.0 | 40.7 | 43.0 | 41.1 | 53.8 | 48.5 | 45.0 | 55.6 | 51.8 |
| Gemini-1.5 (2024) | 94.8 | 98.4 | 43.6 | 42.4 | 65.3 | 48.7 | 55.6 | 79.5 | 59.8 | 70.0 | 66.7 | 66.0 |

Table 3: **Comparison with state-of-the-art methods on TempCompass.** TempCompass (Liu et al., 2024) assesses the temporal understanding capabilities of video language models across five dimensions Video-STaR improves Video-LLaVA performance on TempCompass by 5%.

| Methods | Dataset size | MSVD-QA | | MSRVTT-QA | | TGIF-QA | | ActivityNet-QA | |
|---|---|---|---|---|---|---|---|---|---|
| | | Accuracy | Score | Accuracy | Score | Accuracy | Score | Accuracy | Score |
| VideoChat (2023b) | 4K | 56.3 | 2.8 | 45.0 | 2.5 | 34.4 | 2.3 | - | 2.2 |
| Video-LLaMA (2023a) | 4K | 51.6 | 2.5 | 29.6 | 1.8 | - | - | 12.4 | 1.1 |
| Video-ChatGPT (2023) | 100K | 64.9 | 3.3 | 49.3 | 2.8 | **51.4** | _3.0_ | 35.2 | 2.7 |
| Video-LLaVA* (2023) | 100K | _69.7_ | _3.9_ | _57.4_ | **3.5** | 46.5 | **3.3** | **43.2** | **3.4** |
| Video-LLaVA$^+$ | 650K | 67.8 | 3.8 | 56.0 | _3.4_ | 46.5 | **3.3** | _42.2_ | _3.3_ |
| Vid-LLaVA$^{Gemini}$ | 128K | 67.2 | 3.9 | 56.9 | 3.4 | 45.5 | 3.4 | 42.6 | _3.3_ |
| Video-STaR | 550K | **71.3** | **4.0** | **58.2** | **3.5** | _47.3_ | **3.3** | **43.2** | _3.3_ |
| Gemini-1.5-pro (2024) | - | 71.6 | _3.9_ | 52.6 | 3.2 | 45.0 | 3.1 | 56.7 | - |

Table 4: **Zero-shot Video QA benchmarks**. As can be seen, many models are approaching Gemini performance - indicating that LMMs may be operating near the noise level on these benchmarks.

## 4.1 EXPERIMENTAL SETTING

**Implementation Details.** We initialize from the Video-LLaVA (Lin et al., 2023) model, which utilizes the Vicuna-7B v1.5 (Chiang et al., 2023). We ran three Video-STaR cycles, and each cycle was initialized with the pre-trained Video-LLaVA weights. We train for one epoch using a $128$ batch size, AdamW optimizer, and a cosine learning rate schedule. The learning rate is $2e-5$ with a $0.03$ warmup ratio. In combination with the generated Video-STaR instruction tuning dataset, we additionally utilized the VideoInstruct-$100K$ (Maaz et al., 2023) and the LLaVA v1.5 instruction tuning datasets (Liu et al., 2023a). Additional details are available in the appendix.

**Baselines.** Besides comparing to Video-LLaVA, we also wanted to evaluate the effect of utilizing additional data and naively adapting the source datasets. Therefore, we utilized simple templates to generate question-answer pairs from the video labels and trained Video-LLaVA on the resulting dataset. We will reference this baseline as Video-LLaVA$^+$. Another baseline for adapting Large Multi-modal Models to novel tasks is model distillation, where a stronger video model - in this work, Gemini $1.5$ pro-vision - is utilized to label/annotate a small set of videos (3500 from each dataset) and used to finetune the models. Unlike Video-LLaVA$^+$, we initialized from the fine-tuned Video-LLaVA model and mixed the Gemini-generated dataset with 50% of Video-LLaVA's original dataset to mitigate forgetting. We will reference this baseline as Vid-LLaVA$^{Gemini}$.

**Evaluation Details.** We evaluate on the following benchmarks; the Zero-shot question-answer (QA) benchmarks: MSVD-QA, MSRVTT-QA, TGIF-QA, and ActivityNet-QA (Xu et al., 2017; 2016; Jang et al., 2017; Heilbron et al., 2015). TempCompass (Liu et al., 2024), a multiple-choice fine-grained QA benchmark. Adapted task performance is evaluated by converting source datasets using simple templates and applying the same evaluation protocol as Maaz et al. (Maaz et al., 2023), producing Kinetics700-QA, STAR-benchmark-QA, and FineDiving-QA. This protocol reports two metrics: accuracy (the percentage of correctly answered questions) and the average score (where ChatGPT rates each response on a scale of 1-5 and calculates the mean of these scores). All evaluations utilize the same GPT model (Wu, 2024) ("gpt-3.5-turbo") to ensure consistent comparisons. Due to cost considerations, 1000 videos were randomly selected from each dataset for Gemini evaluation. The reported values are used on ActivitlyNet-QA.

| Methods | Kinetics700-QA | | STAR-bench-QA | | FineDiving-QA | |
|---|---|---|---|---|---|---|
| | Accuracy | Score | Accuracy | Score | Accuracy | Score |
| Video-LLaVA | 50.0 | 3.2 | 24.9 | 2.6 | 17.1 | 2.2 |
| Video-LLaVA$^+$ | 49.5 | 3.2 | 28.8 | 2.8 | 17.6 | 2.2 |
| Vid-LLaVA$^{Gemini}$ | 49.4 | 2.7 | 29.3 | 2.6 | 16.5 | 2.1 |
| Video-STaR | 59.9 | 3.5 | 33.0 | 2.9 | 21.7 | 2.3 |

Table 5: **Adapted Dataset Performance.** Performance metrics on test sets of Kinetics700, Fine-Diving, and STAR-benchmark datasets via converting them to QA following Maaz et al. (2023). Video-STaR shows significant improvement over Video-LLaVA and Video-LLaVA$^+$, showing the potential of Video-STaR for LMM adaptation to new tasks.

## 4.2 QUANTITATIVE EVALUATION ON ZERO-SHOT BENCHMARKS

To evaluate Video-STaR's effect on general video question answering, we evaluated its effect on Video-LLaVA's performance on TempCompass, see Tab. 3. On TempCompass, Video-STaR outperformed Video-LLaVA across the board– by $\sim 5\%$. To see if this performance boost is simply a factor of training on a larger dataset, we also evaluated Video-LLaVA$^+$. Video-LLaVA$^+$ was trained on even a larger video dataset by naively utilizing video labels, and yields a more modest improvement of $3\%$, showing the utility of Video-STaR. TempCompass is also a fine-grained dataset that would be sensitive to hallucinations, indicating that Video-STaR is not more prone to hallucinations compared to existing methods. Gemini 1.5 pro scored an impressive 66.0 on TempCompass, showing there is still much room for improvement on this benchmark.

We then continued and evaluated Video-STaR's effect on zero-shot video QA performance on the MSVD-QA, MSRVTT-QA, TGIF-QA and ActivityNet-QA benchmarks. As can be seen in Tab. 4, Video-STaR achieves performance improvements where, for instance, on the MSVD-QA dataset, Video-STaR attains the highest accuracy of 71.3% vs Video-LLaVA's 69.7. On MSRVTT-QA, Video-STaR leads with an accuracy of $58.2\%$ and maintains a competitive edge in other datasets like TGIF-QA and ActivityNet-QA. Seeing the relatively small performance gains compared to TempCompass, we additionally evaluated Gemini 1.5 pro-vision on 1000 video subsets of each dataset and found that its performance is on par with existing open-source models. We believe this shows that we are near the 'noise' limit of these benchmarks. Our qualitative analysis indicated that many of the questions selected as 'wrong' are actually due to the benchmark design—overly general questions with multiple correct answers. Concurrent work (Wu, 2024) has similarly concluded that the ChatGPT-3.5 version utilized in evaluation can lead to variations of $\pm 10$ in accuracy.

## 4.3 QUANTITATIVE EVALUATION ON ADAPTED DATASETS

Besides improving general visual question-answering performance, Video-STaR can also adapt Large Multi-modal models to novel takes. To demonstrate this, we converted the test sets (*not* included in training) of the source datasets – Kinetics700, STAR-benchmark, and FineDiving. The results of these evaluations are reported in Tab. 5. Adapting LMMs with easier-to-collect labels can be helpful in various applications, leading to a more versatile, multi-domain capable assistant. When evaluating Video-STaR's impact on LMM performance on the diverse source datasets, we found that it significantly improves model performance, particularly on complex tasks. For instance, on Kinetics700, known for its extensive action categories, Video-STaR enhanced Video-LLaVA's performance accuracy by an average of 20% (as can be seen in Tab. 5), showcasing its ability to develop generalized models adept across multiple domains. Interestingly, Video-LLaVA$^+$'s performance did not improve compared to Video-LLaVA, and in some cases, even worsened, showing that one cannot directly utilize labeled datasets for LMM adaptation.

Action Quality Assessment (AQA) is a complex video task requiring detailed action understanding, where Video-STaR significantly enhanced LMM performance on the FineDiving dataset. Our results show a notable improvement from 17.6 to 21.6 in score prediction accuracy, highlighting Video-STaR's effectiveness in refining LMM's temporal reasoning. However, Video-STaR allows LMMs to not only rate a particular dive but also explain the rationale behind each assessment. This rationale is invaluable for many applications, effectively providing potential user feedback for improvement.

| Ablations | Kinetics700-QA | | STAR-bench-QA | | FineDiving-QA | |
|---|---|---|---|---|---|---|
| | Accuracy | Score | Accuracy | Score | Accuracy | Score |
| Video-STaR | **59.9** | **3.5** | **33.0** | **2.9** | **21.6** | **2.3** |
| - Generation | 58.8 | **3.5** | 28.7 | 2.7 | 17.1 | 2.1 |
| - Rationalization | 59.8 | **3.5** | 26.6 | 2.7 | 12.8 | 2.0 |
| - Generation | 50.0 | 3.2 | 24.9 | 2.6 | 17.6 | 2.2 |

Table 6: **Ablations on Adapted Datasets.** Performance metrics on test sets of Kinetics700, STAR-benchmark, and FineDiving datasets. Label Rationalization impacts mostly the difficult datasets, such as FineDiving, whose initial Answer Generation yields are low.

| Ablations | MSVD-QA | | MSRVTT-QA | | TGIF-QA | | ActivityNet-QA | |
|---|---|---|---|---|---|---|---|---|
| | Accuracy | Score | Accuracy | Score | Accuracy | Score | Accuracy | Score |
| Video-STaR | **71.3** | **4.0** | **58.2** | **3.5** | 46.8 | 3.3 | 42.2 | 3.3 |
| - Generation | 70.6 | 3.9 | 57.8 | **3.5** | 44.9 | 3.3 | 41.1 | 3.2 |
| - Rationalization | 70.6 | 3.9 | 57.5 | **3.5** | **47.7** | **3.4** | 42.2 | 3.3 |
| - Generation | 69.7 | 3.9 | 57.4 | **3.5** | 46.5 | 3.3 | **43.2** | **3.4** |

Table 7: **Ablations on Zero-Shot Benchmarks**. In simpler benchmarks, Answer Generation proved more critical for zero-shot generalization than Label Rationalization.

This advancement enables novel applications, from sports coaching to automated feedback systems, by offering evaluations and constructive feedback. The ability to interpret and improve action quality underscores the potential of Video-STaR, underscoring the potential of utilizing LMMs as intelligent and informative visual assistants. For more, please see App. Sec. B.

## 4.4 ABLATIONS

In our ablation studies, we evaluated the impact of removing Label Rationalization and Answer Generation from Video-STaR, focusing on adapted datasets (Kinetics700, FineDiving, STAR-benchmark) and zero-shot benchmarks (MSVD-QA, MSRVTT-QA, TGIF-QA, ActivityNet-QA).

**Adapted Datasets**  For adapted datasets (Tab. 4.3), excluding Label Rationalization led to a significant performance drop in FineDiving, from 21.6 to 12.8 in accuracy, highlighting its critical role in complex reasoning tasks. This is likely due to the lack of conversion of any examples from the data. However, the removal of Answer Generation resulted in a more pronounced and uniform decline across all datasets. For example, Kinetics700's accuracy was reduced from 59.9 to 50.0, underscoring its foundational role in generating context-relevant responses.

**Zero-shot benchmarks**  In zero-shot benchmarks (Tab. 4.3), the removal of Label Rationalization had a mixed impact, slightly affecting MSVD-QA where accuracy decreased from 71.3 to 70.6. The elimination of Answer Generation consistently lowered performance, such as a decrease in MSRVTT-QA accuracy from 58.2 to 57.4. ActivityNet-QA performance improved, probably because $100K$-Instruct utilizes ActivityNet for instruction tuning. Therefore, the introduction of additional videos decreases performance.

## 5 RELATED WORKS

In Sec. 5.1, recent advancements in Large Vision-Language Models and video instruction tuning datasets are introduced. In Sec. 5.2, advancements in Large Language Models and self-supervised instruction tuning are explored.

## 5.1 LARGE VISION-LANGUAGE MODELS

Initial LMMs, such as LLaVA (Liu et al., 2023b;a) and BLIP-2 (Li et al., 2023a), demonstrated the potential of merging image inputs with large language models. Methods like mPLUG-Owl (Ye

et al., 2023) and Flamingo (Alayrac et al., 2022) further allowed for multiple image inputs without architectural changes. Li et al. (2023b) and Zhang et al. (2024) led the transition to video understanding, integrating video/image encoders and LLMs while training on small video instruction tuning datasets. Jin et al. (2023) introduced Chat-UniVi, a unified model employing dynamic visual tokens for both images and videos, optimizing visual token usage and higher frame count sampling. LLaMA-VID (Li et al., 2023c) showed that the token count can be further reduced by pooling the tokens selectively via the text prompt using Q-Former. Wang et al. (2024b) introduced a training-free agentic approach. Recently, Video-LLaVA (Lin et al., 2023) used modality-specific encoders for video and image inputs to leverage LanguageBind encoders as they are constructively aligned during pretraining and utilized a shared projection.

Maaz et al. (2023) expanded the field with the first large video instruction tuning dataset, VideoInstruct-$100K$. VideoInstruct-$100K$ was generated from ActivityNet (Heilbron et al., 2015) by prompting chatGPT with the video captions, generating question-answer pairs. While driving much of the performance improvement in the field (Jin et al., 2023; Wang et al., 2023; Lin et al., 2023; Li et al., 2023c), upon examination of VideoInstruct-$100K$, it is evident that it suffers from quality issues. The questions often degrade into de facto prompts for a video caption (see Fig. 10) and rarely require many spatiotemporal capabilities, which may limit LMM performance.

## 5.2 LARGE LANGUAGE MODELS AND SELF-TRAINING

The advent of GPT (Radford et al., 2018; Brown et al., 2020) marked significant milestones in natural language processing, showcasing LLMs' power in understanding and generating human-like text. Open-source LLMs like LLaMA (Touvron et al., 2023a;b) and their instruction-tuned variants like Alpaca and Vicuna (Taori et al., 2023; Chiang et al., 2023) further tailored these models for nuanced human-AI interactions. However, even LLMs have found it challenging to scale annotated datasets for training, prompting work on self-training and self-improvement (Singh et al., 2023; Huang et al., 2022; Ho et al., 2023; Marasović et al., 2022; Hosseini et al., 2024). In this line of work, LLMs cycle between instruction-tuning data generation and instruction tuning, iteratively improving LLM performance over cycles. For instance, Zelikman et al. (2022) introduced the the Self-Taught Reasoners method, used rationalization to generate chain-of-thought (CoT) reasoning, filtering poor rationalizations to retain correctly answered questions. Other self-training approaches include expectation-maximization-based approaches (Singh et al., 2023), which alternate between data generation and improvement between training cycles. Alternatively, majority voting has also been utilized to generate answers and rationale for unlabeled questions (Huang et al., 2022). These methods show the effectiveness of iterative self-training. In our work, we aim to introduce a weakly supervised self-training approach for video instruction tuning, leveraging video supervision that is often easier to collect and exists in many large and diverse datasets.

## 6 CONCLUSIONS

In conclusion, Video Self-Training with augmented Reasoning (Video-STaR) presents a novel approach to enhance Large Multi-Language Models (LMMs) by enabling the use of diverse labeled video datasets for visual instruction tuning. Video-STaR addresses critical data diversity and quality challenges, leading to performance improvements across various video understanding tasks. Our experiments demonstrate Video-STaR's effectiveness in source dataset adaptation and zero-shot generalization, showcasing its potential in advancing LMM capabilities for complex video reasoning.

The promising results of Video-STaR open new research avenues, particularly in expanding LMM knowledge bases using readily available image and video datasets. Future work could explore advanced self-training techniques and integration with emerging LMM architectures, focusing on long-form video understanding to boost LMM understanding. Additional work is also needed to reduce hallucinations, perhaps by using grounded VLMs as agents to further supervise the generation.

**Acknowledgements.** We thank Google Research for providing financial support through a Stanford HAI–Google collaboration. This work was partially supported by the National Science Foundation under Grant No. 2026498. We thank the Knight-Hennessy Scholars Foundation for funding OZ.

REPRODUCIBILITY STATEMENT

All code, source files, and generated dataset text instructions can be found in our supporting information and will be made publicly available. Additional implementation details can also be found in the Appendix, Sec. A.

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
