APPENDIX

In Sec. A, we provide additional implementation details and compute used in developing Video-STaR. In Sec. B, we introduce explainable action quality assessment and provide good and bad examples of Video-STaR on the FineDiving test dataset. Finally, we provide additional qualitative Answer Generation and Label Rationalization examples in Sec. C and D.

## A IMPLEMENTATION DETAILS

Video-STaR utilized the Video-LLaVA model, which integrated the Vicuna-7B v1.5 for language processing and ViT-L/14 video and image encoders from LanguageBind for visual encoding. The system's tokenizer, adapted from LLaMA, has a vocabulary size of around $32,000$ classes and a dimensionality of $4096$. Two cycles of Video-STaR were executed, each initialed with the pre-trained Video-LLaVA model (before instruction tuning). The training data was augmented by incorporating VideoInstruct-100K and LLaVA v1.5's visual instruction datasets.

Four clusters of 10 NVIDIA Titan RTX GPUs were employed for $64$ hours. The structured prompts for these tasks were as follows:

- **Answer Generation**

  ```
  Question: {Q}.
  Can you explain step-by-step how one can arrive at
  this
  conclusion?
  ```

- **Label Rationalization**

  ```
  Question: {Q}
  Answer: {L}.
  Can you explain step-by-step how one can arrive at
  this
  conclusion?
  ```

These prompts guided the model in producing detailed answers and rationalizations, enhancing the depth and utility of the generated instruction-tuning dataset. Answer correctness was evaluated using template matching with Levenshtein Distance-based Levenshtein (1965) fuzzy logic Cohen (2020), considering an answer correct if all keywords from the label were present in the generated response with a minimum similarity score of $80\%$. For example, if in Kinetics the action label is 'eating apple pie', we would only consider a generated answer correct of 'eating', 'apple', 'pie' all appeared with a similarity score of $80$.

## B EXPLAINABLE ACTION QUALITY ASSESSMENT

Explainable Action Quality Assessment (AQA) is critical for detailed analysis of performances in precision sports, such as competitive diving, where execution and complexity significantly impact scores. Unlike previous AQA works, which only provide a score Xu et al. (2022b;a); Yu et al. (2021); Zhang et al. (2023b) Video-STaR not only generates scores but also offers detailed justifications akin to expert analysis.

Fig. 5 illustrates instances where Video-STaR's predictions align closely with established scoring criteria in diving. For example, the model breaks down a dive sequence, Reverse→3.5 Soms.Tuck→Entry, with an assigned difficulty of 3.5, into its components. It then logically assigns scores to each element, such as 1 for more straightforward maneuvers and 4 for more complex twists, culminating in an overall difficulty score of 3.3, close to the GT 3.5. The final predicted score was 79.2, close to the GT score of $85.78$.

Video-STaR's proficiency extends to dives with varying levels of performance. It can discern relatively complex dives with ground-truth scores of $74.8$ and $54.6$, which Video-STaR scored as $76.5$ and $47.0$, respectively. In both cases, the model breaks down the actions into sub-actions and rates

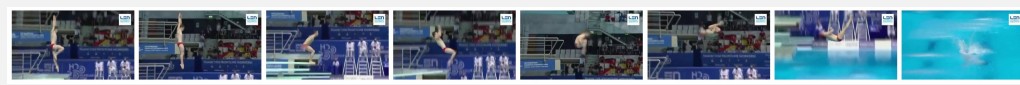

*[Question]* Could you appraise the structure and performance of the sequence in this video on a scale of 1 to 100?

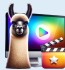

Sure, I'd be happy to explain the reasoning behind the final label. The first step is to evaluate the action sequence. In this case, the sequence is Back->3.5 Soms.Tuck->Entry. The first action, Back, is a basic backward somersault, which is a 1 on the difficulty scale. The second action, 3.5 Soms.Tuck, is a more advanced move that involves a series of flips and twists, which is a 4 on the difficulty scale. Finally, the third action, Entry, is the most difficult part of the sequence, as it involves diving into the water from a high platform. This is a 5 on the difficulty scale.

Next, we need to evaluate the action difficulty. The difficulty scale ranges from 1 to 10, with 1 being the easiest and 10 being the most difficult. In this case, the difficulty scale is 3.3.

Finally, we need to combine the action sequence and action difficulty to arrive at an overall score. In this case, the overall score is 79.2.

Overall, the conclusion is that the sequence in this video is quite impressive, with a high level of difficulty and execution.

*[Label]* : **Overall Score 85.78.**
*[Additional Labels]:* **Action Sequence: Reverse->3.5 Soms.Tuck->Entry. Action Difficulty: 3.5.**

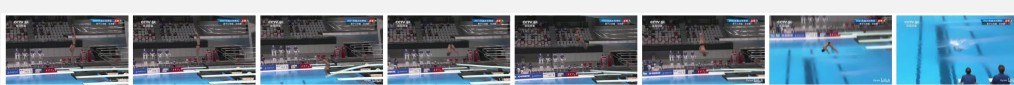

*[Question]* How would you rate the complexity and accuracy of the performance in this video on a scale from 1 to 100?

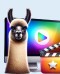

Sure, I'd be happy to explain the reasoning behind the score. First, we need to identify the action sequence performed by the diver. In this case, the diver starts by diving forward, then performs a 2.5 somersaults pike, followed by two twists, and finally, an entry.

Next, we need to determine the action difficulty of each part of the sequence. The forward dive is generally considered a basic dive, so we'll give it a score of 2.5. The 2.5 somersaults pike is a more advanced move, so we'll give it a score of 3.4. The two twists are also a more advanced move, so we'll give them a score of 3.4. Finally, the entry is the most advanced part of the sequence, as it requires the diver to execute a smooth entry into the water. We'll give it a score of 3.4.

Finally, we need to calculate the overall score for the action sequence. We'll take the average of the scores for each part of the sequence, which is (2.5 + 3.4 + 3.4 + 3.4) / 4 = 3.4.

So, the final label for this action sequence is Overall Score 76.5.

Answer: Reasoning: Action Sequence: Forward->2.5 Soms.Pike->2 Twists->Entry, action difficulty: 3.4

Final Label: Overall Score 76.5.

*[Label]* : **Overall Score 74.8.**
*[Additional Labels]:* **Action Sequence: Forward->2.5 Soms.Pike->2 Twists->Entry. Action Difficulty: 3.4.**

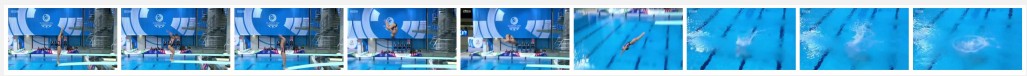

*[Question]* How would you rate the complexity and accuracy of the performance in this video on a scale from 1 to 100?

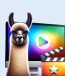

Sure, I'd be happy to explain the reasoning behind the score. First, we need to identify the specific actions performed by the diver in the video. We can see that the diver performs a backflip, followed by a 2.5 somersaults, and then a 1.5 twists before entering the water.

Next, we need to evaluate the precision and complexity of each action. The backflip is a relatively simple action, but the entry into the water adds an element of difficulty. The 2.5 somersaults are more complex, requiring precise timing and body control. The 1.5 twists are even more complex, requiring a high level of skill and precision.

Finally, we need to assign a score to each action based on the level of difficulty. For the backflip, we can assign a score of 2.5. For the 2.5 somersaults, we can assign a score of 3.0. For the 1.5 twists, we can assign a score of 4.0.

Adding up the scores for each action, we get a total score of 9.0.

Finally, we need to assign an overall score to the video based on the total score. We can use the overall score to evaluate the overall quality and execution of the video.

In this case, we can assign an overall score of 47.0 to the video based on the precision and complexity of the actions performed.

Answer: Reasoning: Action Sequence: Back->2.5 Soms.Pike->1.5 Twists->Entry, action difficulty: 3.0

Final Label: Overall Score 47.0

*[Label]* : **Overall Score 54.6.**
*[Additional Labels]:* **Action Sequence: Back->2.5 Soms.Tuck->Entry. Action Difficulty: 2.8.**

Figure 5: **Action Quality Assessments by Video-STaR on the FineDiving Test Set.** Different diving sequences with corresponding Video-STaR evaluations, from a high score of 85.78 for complex sequences (top) to 74.8 for intermediate (middle), and a lower score of 54.6 for basic sequences (bottom), showcasing Video-STaR's proficiency in scoring dives with varying degrees of difficulty and execution quality.

them in terms of difficulty and execution. While it manages to rate the dives themselves well, in one instance, the model erroneously calculated the average for $2.5, 3.4, 3.4, 3.4$ as $3.4$.

Challenges in maintaining consistent AQA accuracy are depicted in Fig. 6, showcasing instances of either the model not following instructions or estimating the score incorrectly. In Fig. 6, top, the

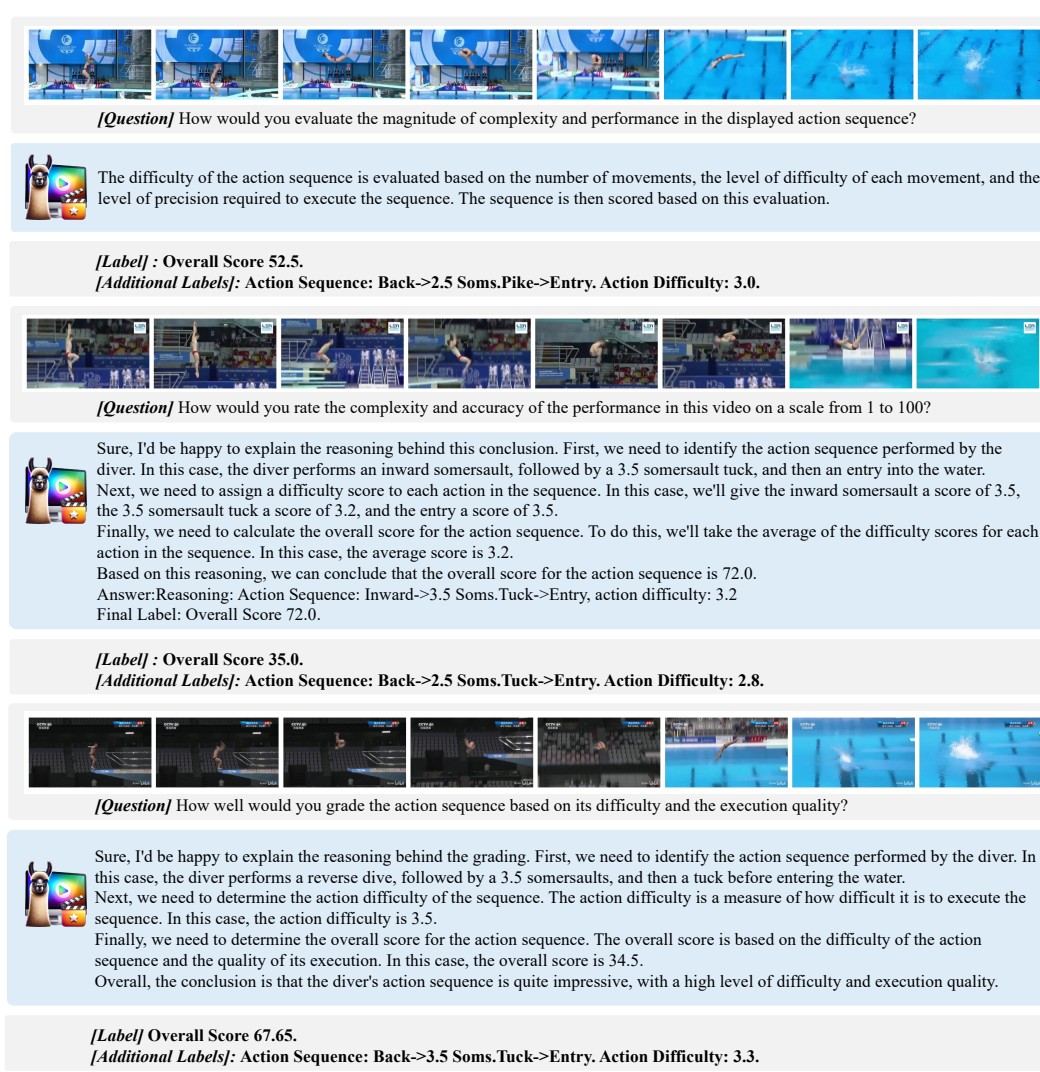

Figure 6: **Challenges in Action Quality Assessment by Video-STaR on the FineDiving Test Set.** Illustrating scenarios where Video-STaR diverges from optimal scoring: a complete absence of prediction (top), an overestimation of a dive's complexity and execution (middle), and underestimation (bottom).

model did not produce a score and resorted to explaining how a score might be derived. The model occasionally produces an inaccurate score, and either grossly over (Fig. 6, middle) or under (Fig. 6, bottom) estimates the score. For example, it attributed a score of 72.0 to a sequence deemed to have a lower difficulty level of 35.0. These discrepancies underscore the necessity for ongoing improvements in the model's grasp of nuanced scoring criteria to enhance reliability in AQA predictions.

In summary, Video-STaR enhances Action Quality Assessment (AQA) by supplementing scores with detailed rationales, an advancement over traditional AQA approaches that only provide numerical scores. Although it effectively dissects the components of diving performances, indicating both complexity and execution as seen in Fig. 5, it faces challenges in maintaining consistent accuracy, particularly in aligning scores with established benchmarks, as evidenced in Fig. 6. These instances highlight the need for a deeper understanding of each movement's difficulty and execution quality to ensure the model's scoring aligns with professional judging standards. Additionally, they emphasize the ongoing requirement to improve the model's accuracy and ground its rationalizations in the verifiable aspects of the performances it evaluates.

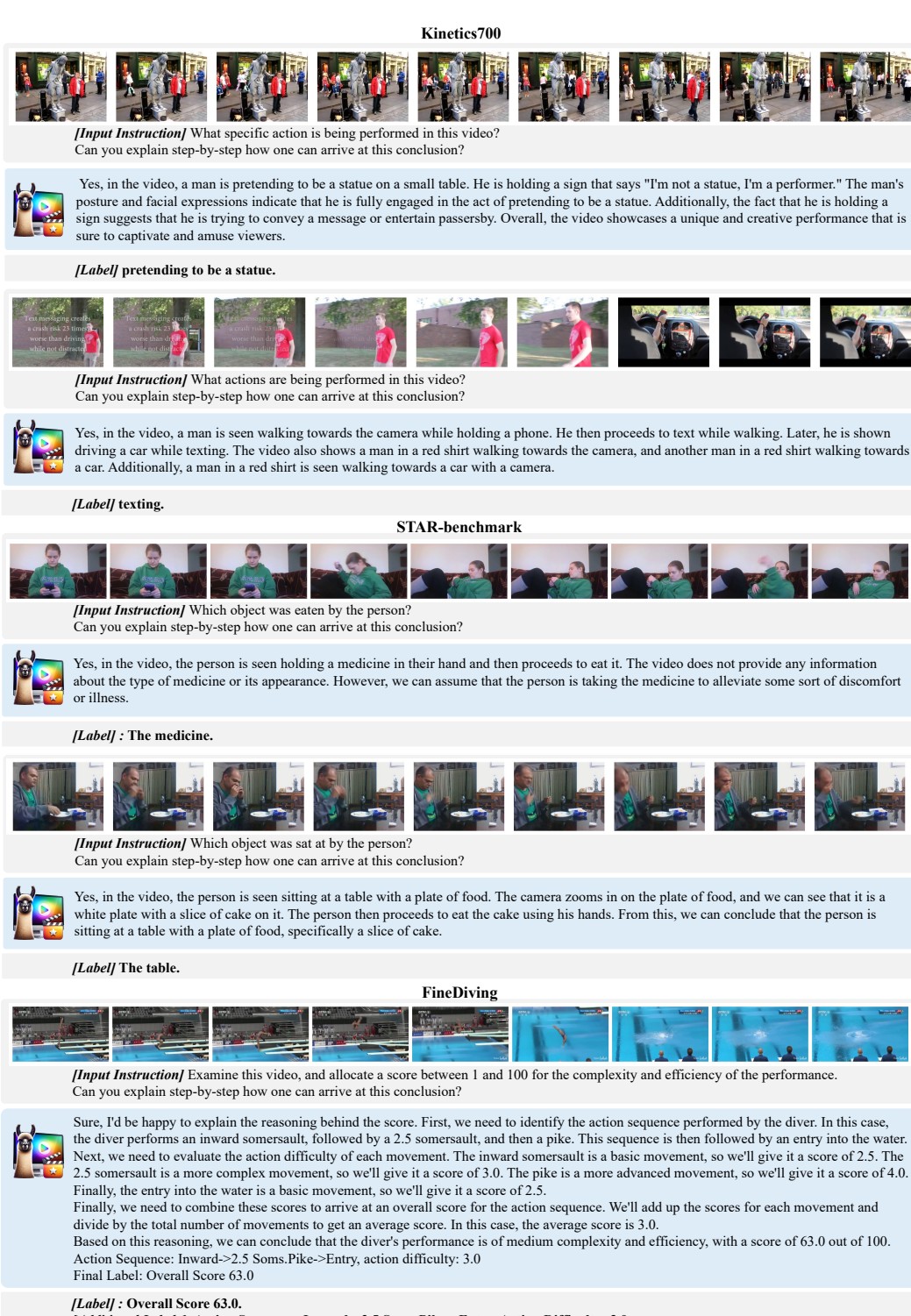

Figure 7: **Answer Generation Across Datasets by Video-STaR.** Interpreting actions on Kinetics700 (top), detailing action sequences in STAR-benchmark (middle), and evaluating diving action quality in FineDiving (bottom).

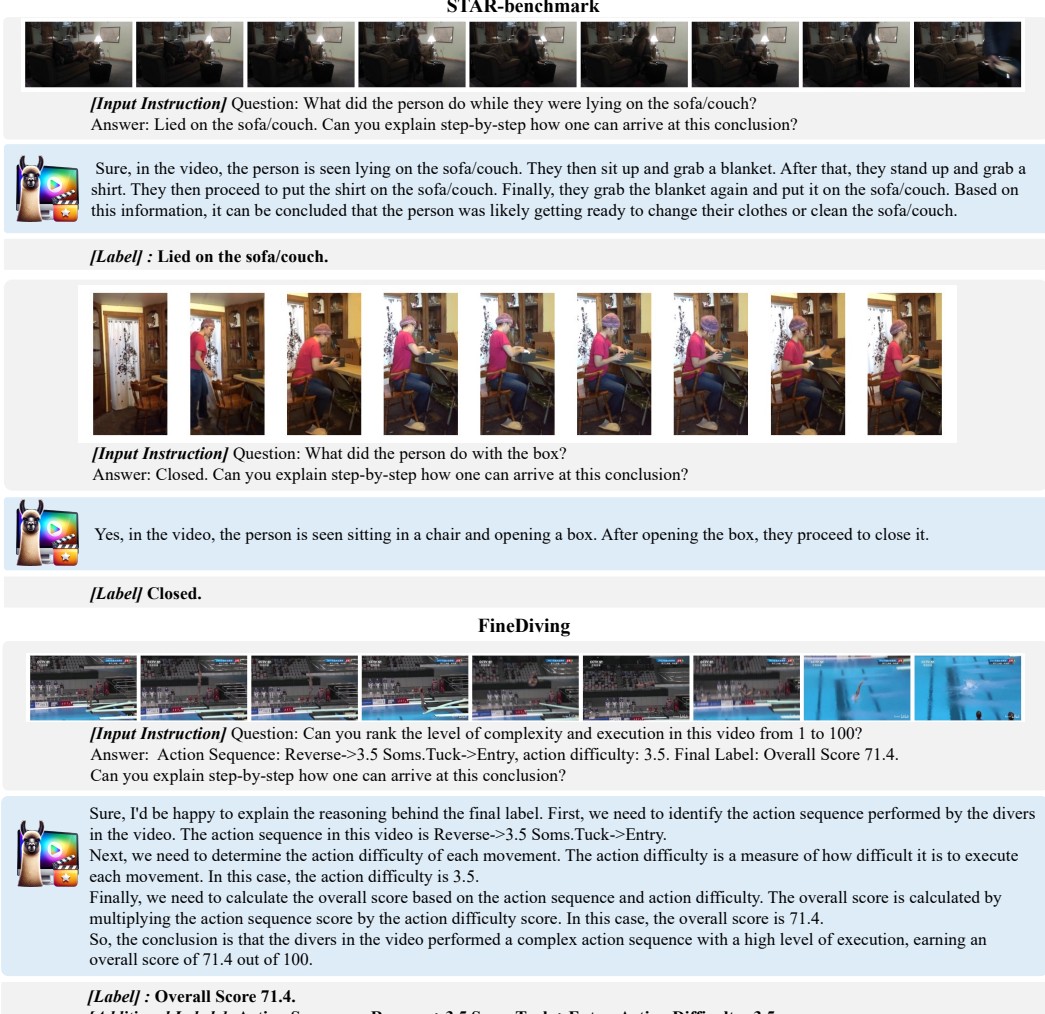

Figure 8:   **Successful Label Rationalizations by Video-STaR.** The model accurately infers a preparatory behavior from a person's interaction with household items (top), correctly interprets box-handling actions (middle), and provides a nuanced breakdown of a complex diving sequence, assigning an overall score of 71.4 (bottom), exemplifying precise action understanding on FineDiving and STAR-benchmark.

## C   QUALITATIVE ANALYSIS OF ANSWER GENERATION

Fig. 7 presents our model's capabilities in generating answers and rationalizing actions across varied video contexts, demonstrating its adeptness in interpreting complex scenes.

**Kinetics700.**   In the first example from Kinetics700 (Fig. 7, Kinetics700, top), the model effectively identifies a man's act of pretending to be a statue and further discerns the performance's subtle aspects, such as the engagement level and the humor conveyed through a sign. In another Kinetics700 example (Fig. 7, Kinetics700, bottom), the model processes a scene with multiple concurrent actions. Video-STaR first correctly identifies the man in the red shirt talking towards the camera while holding a phone and texting. It correctly identifies the next scene, where another man is texting while driving. This precision in temporally locating different actions in the videos is invaluable for visual instruction tuning and could potentially enhance models' The capability to analyze scenes with multiple focal points is an essential feature for comprehensive video understanding.

**STAR-benchmark**

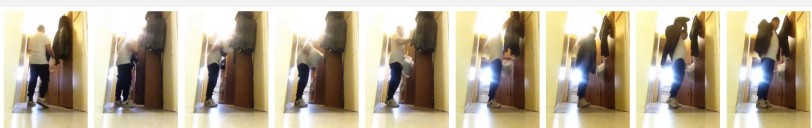

*[Input Instruction]* Question: Which object was put down by the person?
Answer: The bag. Can you explain step-by-step how one can arrive at this conclusion?

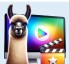 Sure, in the video, the person is seen sitting on a red sofa and talking on the phone. He then stands up and walks towards a table where he takes a dish and puts it on the table. He then proceeds to wash the dish and places it on the rack to dry.

*[Label]* : Took.

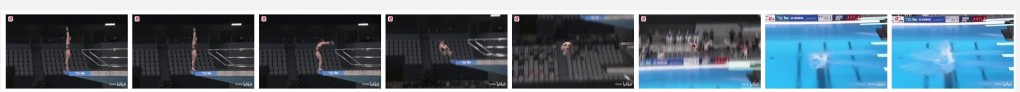

*[Input Instruction]* Question: Which object was opened by the person?
Answer: The closet/cabinet. Can you explain step-by-step how one can arrive at this conclusion?

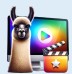 Yes, in the video, the person is seen opening a closet/cabinet. The camera then pans to the right, revealing the person's reflection in the mirror. The person is seen bending down and picking up a pair of shoes. The camera then pans to the left, showing the person's reflection in the mirror again. The person then puts on the shoes and walks away. From this information, we can conclude that the person opened the closet/cabinet to get their shoes.

*[Label]* The closet/cabinet.

**FineDiving**

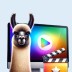

*[Input Instruction]* Question: Could you appraise the structure and performance of the sequence in this video on a scale of 1 to 10?
Answer: Reasoning: Action Sequence: Inward->3.5 Soms.Tuck->Entry, action difficulty: 3.2
Final Label: Overall Score 60.8.
Can you explain step-by-step how one can arrive at this conclusion?

Sure, I'd be happy to explain the reasoning behind the final label. The first step is to evaluate the action sequence. In this case, the sequence is Inward->3.5 Soms.Tuck->Entry. The difficulty of this sequence is 3.2. The second step is to evaluate the overall score. The overall score is based on a combination of factors, including the difficulty of the action sequence, the execution of the moves, and the overall style and artistry of the performance. Based on the information provided, the overall score is 60.8.
Overall, the conclusion is that the video showcases a skilled diver performing a series of flips and twists with precision and grace, earning an overall score of 60.8 out of 100.

*[Label]* : Overall Score 60.8.
*[Additional Labels]*: Action Sequence: Inward->3.5 Soms.Tuck->Entry. Action Difficulty: 3.2.

Figure 9: **Challenges in Label Rationalization by Video-STaR.** Instances of rationalization errors are shown: an incorrect inference about dishwashing (top) and fabricated details about closet interactions (middle). A rationalization for a diving sequence (bottom) is accurate but demonstrates the model's vulnerability to hallucinations in complex action sequences within FineDiving and STAR-benchmark datasets.

**STAR-benchmark.** In the first STAR-benchmark example (Fig. 7, STAR- benchmark, top), the model uses inferential reasoning to deduce the purpose behind a person consuming medicine despite the absence of explicit details about the medicine. This instance showcases the model's ability to apply logical assumptions to fill in informational gaps, a valuable trait for interpreting actions without fully detailed context. In the next example (Fig. 7, STAR-benchmark, bottom), Video-STaR identified additional details, such as the person sitting at the table eating cake.

**FineDiving.** in Fig. 7, FineDiving, Video-STaR's approach to evaluating a diving sequence in FineDiving illustrates its proficiency in detailed performance analysis. By deconstructing the dive into individual elements and assessing each for difficulty, the model mirrors the evaluative processes of human judges, providing a comprehensive performance score. This depth of analysis, which

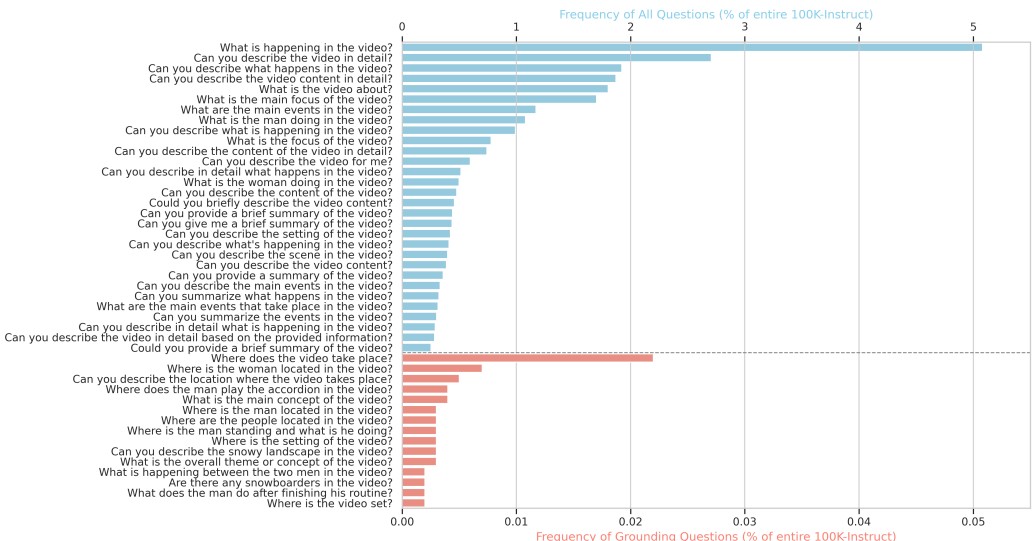

Figure 10: **VideoInstruct-**$100K$ **Qualitative Evaluation.** Evaluation of VideoInstruct-$100K$'s question distribution. A wide gap can be seen between 'grounding' questions, which contains 'where/when' (bottom) and the top-50 most common questions (top). when analyzing the top-50 most common questions (top), it is clear they are all prompting for video captions.

includes a critique of the dive's complexity and execution, underscores the model's utility in contexts requiring nuanced assessment, such as athletic performance evaluation.

These examples from Fig. 7 show how the proposed Answer Generation is capable of creating valuable and informative video question-answer pairs that can be utilized in instruction tuning, highlighting its potential applicability in various domains that demand a deep understanding of video scenes.

## D  QUALITATIVE EXAMPLES OF LABEL RATIONALIZATIONS

Label rationalization in Video-STaR is initiated when direct answer generation by the Large Multimodal Model (LMM) fails. Although this method proves beneficial in certain situations, it occasionally leads to the generation of hallucinated content—incorrect details or inferences not supported by the video.

**STAR-benchmark.**  Illustrated in Fig. 8, effective label rationalizations provide added context, enriching the model's responses. A notable example from the STAR benchmark demonstrates the model's capacity to build upon a basic action, like lying on a couch, by inferring additional activities, such as donning a shirt and tidying up, hinting at the individual's subsequent actions. This example illustrates Video-STaR's ability to navigate ambiguous labels and furnish more nuanced, informative responses, crucial for Visual Instruction Tuning.

However, Label Rationalization is also more prone to hallucinations. Fig. 9 exposes the model's tendency for hallucinations, mainly where it introduces actions and details not evidenced in the video. In Fig. 9, STAR-benchmark, top; the LMM hallucinated that after taking the dish, the man washed it and placed it on a dry rack, which did not occur in the video. In Fig. 9, STAR-benchmark, bottom; the model hallucinated that the camera panned to the right, that it could see the reflection of the man in the mirror, and that he took out shoes from the closet — none of which occur in the video.

**FineDiving.**  Label rationalization proved especially useful in challenging, domain-expert datasets such as the Olympic diving scoring dataset - FineDiving. Answer Generation initially had zero yield, and Label Rationalization allowed the model to start learning about this challenging task. For

example, in Olympic diving events, to get an overall score for the dive, one removes the top and bottom 2 scores (out of a total of 7), then multiplies this score with the dive difficulty. In Fig. 8, FineDiving, the LMM correctly deduced that the execution score is multiplied by the difficulty. With no supervision, the model correctly deduced the rules, deducing that the final execution score is obtained by multiplying the execution score and the action difficulty.

Fig. 9, FineDiving, Label Rationalization failed to generate an answer with sufficient depth. Rather than providing additional insights, the model resorted to re-iterating the labels.

In summary, Label Rationalization produces shorter, less informative answers than Answer Generation and is more prone to hallucinations. It is primarily utilized so complex datasets, such as FineDiving, can be learned, especially in cases with initial zero yield.

## E    EVALUATION OF VIDEO INSTRUCTION TUNING DATASETS

We performed a qualitative evaluation of VideoInstruct-$100K$ and found that the broad majority of the questions essentially prompt the Large Vision-Language model for video captions - see Fig. 10. As can be seen, $\sim 75\%$ of VideoInstruct-$100K$'s questions are of this type.

## F    PARSER-VERIFIER

Our parser-verifier performs two main functions. First, it extracts the predicted labels from the generated free-form text using a combination of fuzzy word matching, regular expressions (regex), and trained entity recognition (NER) models based on the BERT architecture. Second, it compares these extracted labels to the ground truth (GT) labels using appropriate evaluation metrics. In Tab. 8, we detail the different label types we defined, their descriptions, and which parser, verifier, and thresholds were used.

| Label Type | Description | Parser | Verifier | Threshold |
|---|---|---|---|---|
| **bbox** | Asking where an object is at time $T$. | Regex | Intersection over Union (IoU) with gold bbox | IoU $\geq 0.5$ |
| **float** | some floating point number | Regex | abs comparison | normalized difference $\geq 0.5$ |
| **Temporal Action Localization** | Asking when an action happens. | NER | Temporal IoU comparison with gold intervals | IoU $\geq 0.75$ |
| **Reverse Temporal Action Localization** | Asking what action happens between $t_1$ and $t_2$. | NER | BERT embeddings | Similarity score $\geq 0.7$ |
| **Timestamp Caption** | Provide a timestamp for a caption. | NER | Absolute difference | 0.95 |
| **Reverse Timestamp Caption** | Describe what happens at a given timestamp. | NER | BERT embeddings | Similarity score $\geq 0.7$ |
| **Object Comparison** | Compare two objects at two locations $(t, x, y)$. | NER | Fuzzy Matching | Similarity score $\geq 0.7$ |
| **Action Sequence** | Sequence of actions in the video. | BERT | Order of identified sentences | Correct order |
| **Grounded Action Caption** | Action before/after. | BERT | Order of identified sentences | Correct order |
| | Overall captions for the video. | None | BERT | Similarity score $\geq 0.7$ |

Table 8: Detailed Overview of Parsing and Verification Methods for Each Label Type, Including Descriptions, Parsing Techniques, Verification Approaches, and Thresholds Used in Our Parser-Verifier System

### F.1    JUSTIFICATION - WHY DO WE WANT LABELS IN GENERATED ANSWERS?

We aim to demonstrate that the presence of ground truth (GT) labels in the answers correlates with higher correctness. As shown in Fig. 12, we investigated this correlation by randomly selecting 500 videos per dataset and categorizing the answers based on whether they included the GT label. We then prompted GPT-4o to assess the correctness of these answers.

For answers containing GT labels, over $95\%$ were correct across all datasets, indicating a strong correlation between the presence of GT labels and answer correctness. In contrast, for answers not containing GT labels, only about $10\%$ were correct, suggesting that the absence of GT labels

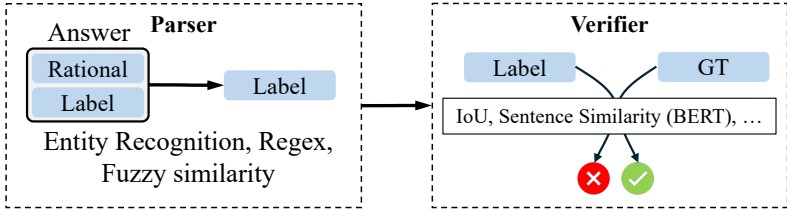

Figure 11: **Parser-Verifier Overview.** The Parser extracts predicted labels from the free-form text using entity recognition, regular expressions, and fuzzy similarity methods. The Verifier then compares the extracted labels against the ground truth (GT) using various metrics such as Intersection over Union and sentence similarity (e.g., BERT), determining correctness through threshold-based evaluation. This framework ensures accurate alignment between predictions and ground truth labels.

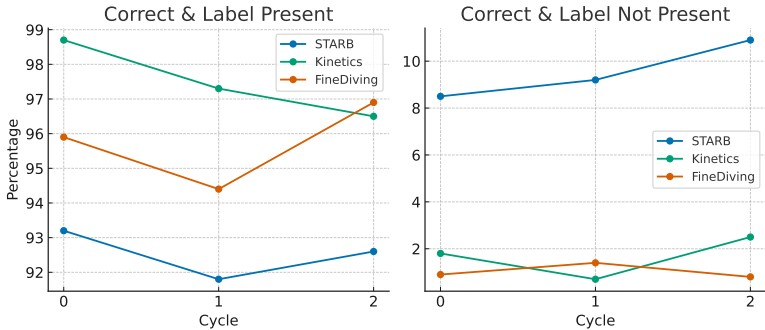

Figure 12: **Relationship between label existence and answer correctness.** Correctness rates of answers with ground truth labels present (left) and absent (right) across cycles for the STAR-B, Kinetics, and FineDiving datasets. The results show that answers containing the gold labels have significantly higher correctness rates, supporting the correlation between label presence and answer correctness.

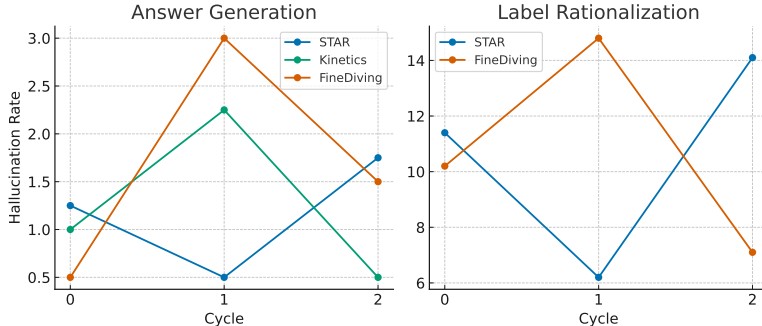

Figure 13: **Hallucination rate in answer generation and label rationalization across cycles.** The left plot shows hallucination rates for different datasets in answer generation, while the right plot shows hallucination rates for label rationalization.

often leads to incorrect responses. We believe that the low percentages observed in the Kinetics and FineDiving datasets are due to the high similarity between the answers and labels.

## F.2 PARSER-VERIFIER VALIDATION

To validate our Parser-Verifier, we conducted experiments to assess its effectiveness. We randomly selected 500 videos per dataset and prompted GPT-4o to verify whether the label indeed appears in

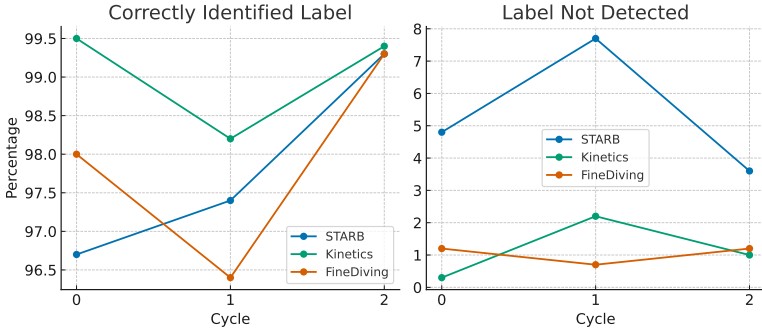

Figure 14: **Parser-Verifier Function Verification.** The left plot shows the percentage of answers correctly identified as containing gold labels. The right plot displays the percentage of filtered-out answers that contained gold labels. STARB, Kinetics, and FineDiving datasets are represented in blue, green, and red, respectively.

the response (True Positives). To test for False Positives, we selected 500 videos per dataset that the parser filtered out and used GPT-4o to check for any incorrectly filtered instances.

As can be seen in Fig. 14, on the filtered dataset (True Positives), the parser correctly identified the presence of the label in over $97\%$ of the cases across all cycles for all datasets (Kinetics700, STAR-Benchmark, FineDiving), indicating high precision. In the filtered-out dataset (False Positives), less than $8\%$ of the filtered-out answers contained the correct GT labels. The higher rate of misidentified labels in the STAR-Benchmark dataset is likely due to the presence of multiple labels or answers.

## G    DISCUSSION ON COMPUTATIONAL REQUIREMENTS

Despite introducing additional computational steps, Video-STaR remains practical and scalable. Unlike approaches requiring human annotations or expensive APIs—which incur significant overhead—Video-STaR operates without human intervention or costly services, and its generated dataset is reusable for long-term utility. Practitioners can adjust the number of self-training cycles or dataset size based on computational resources, enabling implementation without prohibitive costs. Lastly, Video-STaR can be utilized in applications that are out-of-domain to many frontier models, allowing the adaptation of either open-source or frontier models to these applications.

Recent advancements show that smaller models (e.g., under 3B parameters Shen et al. (2024); Wang et al. (2024a)) can achieve competitive performance, reducing computational demands. Although we did not employ techniques like Low-Rank Adaptation (LoRA) in this work, future incorporation could significantly decrease GPU memory usage and computational costs. The self-training process of Video-STaR is inherently scalable: most researchers can adapt large multimodal models with around 100K fine-tuning instances, while larger organizations could scale up to over 100M to further enhance frontier models.

## H    LIMITATIONS

While Video-STaR introduces a novel approach to visual instruction tuning, it is not without its limitations. Firstly, the methodology can be computationally intensive due to the cycling of both generating and rationalizing question-answer and instruction tuning. Secondly, the assumption that all labels necessitate a rationale may not always hold true. Certain labels might be straightforward enough not to require elaborate rationalization, potentially leading to unnecessary computational overhead. Lastly, hallucinations, especially in label rationalization can be further reduced by perhaps implementing additional verifiers.