# OpenReview forum: "Video-STaR: Self-Training Enables Video Instruction Tuning with Any Supervision"
_ICLR.cc/2025/Conference — ICLR 2025 Poster_

### Official Review · Reviewer_ke7n · 2024-11-03

**Soundness:** 2
**Presentation:** 3
**Contribution:** 3
**Rating:** 8
**Confidence:** 3

**Summary:**

The paper proposes a novel method to perform instruction tuning of Large Multi-modal Models (LMMs) on labeled video datasets, where the label format is arbitrary and different from that needed for instruction tuning. The main idea is to use the original LMM itself to generate question-answer pairs for instruction tuning, while using the original dataset labels as a "verifier" to filter out low-quality generations. The authors perform multiple cycles of dataset generation and model training to obtain the final model Video-STAR. Video-STAR shows improved performance on TempCompass and zero-shot video QA datasets.

**Strengths:**

* The idea of turning existing labeled video datasets into instruction tuning dataset and use them for self-training is elegant and effective.
* The paper is well written and is easy to follow
* I appreciate the ablation study

**Weaknesses:**

* The main weakness of the paper is the Video-LLaVA baselines, i.e. Video-LLaVA+ and Video-LLaVA-gemini, which are the main points of comparison in the paper. Video-LLaVA-gemini has been fine-tuned only on several thousand sample pairs, which is incomparable to hundreds of thousands of video samples used for Video-STAR. As for Video-LLaVA+, the number of training samples is comparable, yet there is no information about how those samples were obtained. The way the samples in Video-LLaVA+ are constructed from the original video labels would affect the final model performance a lot, so it is important to explain this in details.
* Additionally, one important baseline for dataset creation is missing. It looks like the SoTA method for creating instruction datasets from video is VideoInstruct [A]. I would expect the authors to compare to this method of dataset creation, both in answer generation and label rationalization settings, to understand how much the video stream actually helps.
* It is somewhat unclear from the paper if Label Rationalization can be used to generate the entire dataset, without using Answer Generation. Somewhere in the paper the authors suggest that it leads to hallucinations, yet in Tables 6 and 7 they show the improvement.

This is a preliminary rating and I will revise my score once the weaknesses and questions are addressed.

[A] -  Maaz et al. "Video-ChatGPT: Towards Detailed Video Understanding via Large Vision and Language Models"

**Questions:**

* In Tables 6 and 7, the last row "- Generation" means that neither generation nor rationalization are used, or only that generation is not used while rationalization is used? Either way, it is important to see both experiments to better understand the contribution of each method.
* When doing self-training in multiple cycles, every time the model is initialized from the same non-instruction-finetuned checkpoint. Why not reuse the model from the previous cycle of self-training as initialization?
* How do the authors deal with numerical labels such as temporal localization, bounding box, performance score? Do they expect the LVM to predict those directly and compute the L1 distance in the verifier? If so, how accurate is the prediction of the model in this case? Is Answer Generation capable of generating good question-answer pairs or Label Rationalization is more helpful here?

---

> ### Author Response · Authors · 2024-11-20
> **[1/3] Response to  Reviewer ke7n**
>
> We sincerely thank Reviewer ke7n for their thoughtful and constructive feedback. We appreciate your recognition of the elegance and effectiveness of our method, as well as your positive comments on our writing and ablation study. In response to your valuable feedback, we have conducted **multiple additional experiments, including training two new models, and have updated the manuscript accordingly.** Your insights have significantly helped us improve the clarity and depth of our paper, and we address your concerns and questions in detail below.
>
> ### 1. Comparison with Video-LLaVA Baselines and Dataset Details
>
> The Video-LLaVA+ was generated using the same questions in Video-STaR but in a method similar to what you discussed in Weakness 2. This experiment aims to show that the performance improvements are not due to the increased dataset size but due to the CoT QA generations.
>
> We understand that Reviewer ke7n feels that our Gemini distillation baseline is too weak, **so we created a more robust comparison and updated our manuscript accordingly**.
>
> **Improved Video-LLaVA^Gemini Baseline:**
> * **Model Initialization:** Instead of starting from the Video-LLaVA pretrained model, we initialized from the SFT variant, which has already seen approximately 100K videos. This provides a stronger baseline.
> * **Increased Labeled Data**: We labeled an additional 8K videos using Gemini, totaling 10K videos for fine-tuning. While labeling more videos is ideal, it is cost-prohibitive (approximately $1,000 USD for 10K videos), limiting our ability to scale this further.
> * **Training Procedure**: We fine-tuned the model on a mixture of 50% Gemini-labeled data and 50% from the original Video-LLaVA mixture to mitigate catastrophic forgetting.
>
> This improved baseline is more competitive and provides a better comparison to our method. The results are shown in the updated tables below.
>
>
>
> Table:  Results of improved baseline (Gemini Distillation) on the adapted datasets
>
> | Methods         |     Kinetics700-QA      | | STAR-bench-QA        | | FineDiving-QA     | |
> |---------------------|----------|-------------|----------|-------------|-----------|----------|
> |                     | Accuracy | Score    | Accuracy | Score    | Accuracy | Score    |
> | Video-LLaVA         | 50.0     | 3.2      | 24.9     | 2.6      | 17.1     | 2.2      |
> | Video-LLaVA+        | 49.5     | 3.2      | 28.8     | 2.8      | 17.6     | 2.2      |
> | *Vid-LLaVA^Gemini* | *49.4*     | *2.7*      | *29.3*     | *2.6*      | *16.5*     |  *2.1*      |
> | **Video-STaR**      | **59.9** | **3.5**  | **33.0** | **2.9**  | **21.7** | **2.3**  |
>
>
> Table:  Results of improved baseline (Gemini Distillation) on TempCompass
>
>
> | Methods              | Action  |       | Direction  |   | Speed  |       | Event    |  Attribute |  |  | | Avg.  |
> |----------------------|------|----------|------|--------|------|------|--------|--------|------|--------|-------|-------|
> |                              | Fine | Coarse  | Obj. | Cam. | Abs. | Rel. | Order | Color | Size | Both | Other |
> | Random               | 39.7 | 40.1     | 39.8 | 39.0 | 40.8 | 42.0 | 41.5  | 40.4  | 39.9 | 38.9 | 39.4  | 40.5  |
> | mPLUG-Owl     | 48.8 | 66.1     | 38.7 | 36.8 | 42.2 | 38.4 | 42.0  | 41.7  | 44.7 | 41.9 | 39.9  | 44.4  |
> | Video-LLaVA    | 63.4 | 93.5     | 36.1 | 34.8 | 42.7 | 26.5 | 39.1  | 52.6  | 37.1 | 43.3 | 33.3  | 45.7  |
> | Video-LLaVA+         | 62.1 | 93.0     | 35.0 | 32.6 | 41.1 | 38.7 | 36.4  | 59.0  | 40.2 | 36.7 | 44.4  | 47.2  |
> | *Vid-LLaVA^Gemini*   | *65.7* | *90.5*  | *38.9* | *46.0* | *41.8* | *42.4* | *41.0* | *48.7* | *49.1* | *32.5* | *40.4* | *48.7* |
> | **Video-STaR**       | **68.6** | **94.1** | **39.9** | **39.0** | **40.7** | **43.0** | **41.1** | **53.8** | **48.5** | **45.0** | **55.6** | **51.8** |
> | Gemini-1.5     | 94.8 | 98.4     | 43.6 | 42.4 | 65.3 | 48.7 | 55.6  | 79.5  | 59.8 | 70.0 | 66.7  | 66.0  |

---

> > ### Author Response · Authors · 2024-11-20
> > **[2/3] Response to  Reviewer ke7n**
> >
> > Table:  Results of improved baseline (Gemini Distillation) on  Zero-shot Video QA benchmarks
> >
> > | Methods              | Dataset Size | MSVD-QA  |         | MSRVTT-QA     |    | TGIF-QA     |     | ActivityNet-QA |   |
> > |---------------|---------|----------|-------|----------|-------|----------|-------|----------|-------|
> > |                      |              | Accuracy | Score | Accuracy | Score | Accuracy | Score | Accuracy | Score |
> > | VideoChat    | 4K           | 56.3     | 2.8   | 45.0     | 2.5   | 34.4     | 2.3   | -        | 2.2   |
> > | Video-LLAMA   | 4K           | 51.6     | 2.5   | 29.6     | 1.8   | -        | -     | 12.4     | 1.1   |
> > | Video-ChatGPT  | 100K         | 64.9     | 3.3   | 49.3     | 2.8   | 51.4     | 3.0   | 35.2     | 2.7   |
> > | Video-LLaVA*   | 100K         | 69.7     | 3.9   | 57.4     | 3.5   | 46.5     | 3.3   | 43.2     | 3.4   |
> > | Video-LLaVA+         | 650K         | 67.8     | 3.8   | 56.0     | 3.4   | 46.5     | 3.3   | 42.2     | 3.3   |
> > | *Vid-LLaVA^Gemini* | *128K*  | *67.2*     | *3.9*   | *56.9*     | *3.4*   | *45.5*     | *3.4*   | *42.6*     | *3.3*   |
> > | **Video-STaR**       | **550K**     | **71.3** | **4.0** | **58.2** | **3.5** | **47.3** | **3.3** | **43.2** | **3.3** |
> > | Gemini-1.5-pro | -            | 71.6     | 3.9   | 52.6     | 3.2   | 45.0     | 3.1   | 56.7     | -     |
> >
> >
> >
> > Overall, this baseline aims to mirror LIMA-style (https://arxiv.org/abs/2305.11206) distillation more closely rather than ultimately generate a large dataset using Gemini. We are confident that, at the right scale, this ultimately will outperform Video-STaR. However, this approach is very expensive, requiring 10s of thousands of API costs, and cannot be used to improve frontier models.
> >
> >
> > These experiments show that while increasing the amount of labeled data with Gemini can improve performance, our method achieves better results without incurring the high labeling costs associated with using large-scale API services like Gemini.
> > Regarding the construction of the Video-LLaVA+ samples, we apologize for not providing sufficient details in the original manuscript. The Video-LLaVA+ dataset was generated using the same questions as in Video-STaR, but without the Chain-of-Thought (CoT) reasoning steps in the answers. The goal was to isolate the impact of CoT and self-training cycles on performance. We will update the manuscript to include detailed descriptions of how these samples were obtained and how they differ from our method.
> >
> > ### 2. Comparison with VideoInstruct (Video-ChatGPT) Method
> > Thank you for pointing out the importance of comparing our method with VideoInstruct [A], which is a state-of-the-art approach for creating instruction datasets from videos. As discussed, Video-LLaVA+ generates the instruction tuning data in a similar approach to [A], where we provided a chatbot with the question, video labels, and available metadata and asked it to generate a response. This differs from [A] as there were no available captions for the videos, but we substituted these for the video labels.
> >
> > ### 3. Ablation tables and utilizing only rationalization
> > Tables 6 & 7 do not have rationalization-only ablations. We acknowledge that it is unclear whether Label Rationalization can generate the entire dataset without using Answer Generation. In our experiments, we found that relying solely on Label Rationalization can lead to higher rates of hallucination compared to using Answer Generation (see revised manuscript, App. Fig. 13). As such, we ran this ablation and report the results below. **This experiment as been added to the ablation tables in the revised manuscript**:
> >
> >
> > | **Ablations**               | **MSVD-QA Accuracy** | **MSVD-QA Score** | **MSRVTT-QA Accuracy** | **MSRVTT-QA Score** | **TGIF-QA Accuracy** | **TGIF-QA Score** | **ActivityNet-QA Accuracy** | **ActivityNet-QA Score** |
> > |--------|--------|--------|--------|-----------|------|-------|----------|----------|
> > | **Video-STaR**              | 71.3             | 4.0           | 58.2               | 3.5            | 46.8             | 3.3           | 42.2                    | 3.3                  |
> > | **- Generation**            | 70.6                 | 3.9               | 57.8                   | 3.5             | 44.9                 | 3.3               | 41.1                       | 3.2                      |
> >
> >
> > | **Ablations**               | **Kinetics700-QA Accuracy** | **Kinetics700-QA Score** | **STAR-bench-QA Accuracy** | **STAR-bench-QA Score** | **FineDiving-QA Accuracy** | **FineDiving-QA Score** |
> > |----------|--------|---------|----------|-------|-------|--------|
> > | **Video-STaR**              | 59.9                    | 3.5                 | 33.0                  | 2.9                 | 21.6                  | 2.3                |
> > | **- Generation**            | 58.8                        | 3.5                  | 28.7                       | 2.7                     | 17.1                      | 2.1                    |

---

> > > ### Author Response · Authors · 2024-11-20
> > > **[3/3] Response to  Reviewer ke7n**
> > >
> > > ## Questions:
> > > ### 1. Ablation of answer generation.
> > > You are correct.
> > > That row indicates no generation or rationalization. As requested by Reviewer ke7n, we trained a model using only label rationalization and updated Tables 6 and 7. Results are presented and discussed in Weakness 3 above. We will update the manuscript to make this clarification explicit and to include the additional experimental results.
> > >
> > > ### 2. Checkpoint initialization
> > > We initiated from the pre-trained checkpoint as this prevents the accumulation of worse-quality data/errors, which is more dominant in the first training cycles, as a larger portion of the dataset is generated via label rationalization rather than answer generation. Initializing from the pre-trained checkpoint mitigates this concern, and is common practice in LLM self-training:
> > >
> > > [1] https://arxiv.org/pdf/2401.10020
> > >
> > > [2] https://arxiv.org/pdf/2408.02666
> > >
> > > [3] https://arxiv.org/pdf/2212.10560
> > >
> > > [4] https://arxiv.org/abs/2203.14465
> > >
> > > As well as more traditional self-improvement approaches
> > > https://ai.stanford.edu/~koller/Papers/Kumar+al:NIPS10.pdf
> > >
> > > ### 3. How are numerical labels verified?
> > > You are correct that we expect the model to predict numerical labels directly, such as timestamps, bounding box coordinates, and performance scores. To make this more manageable for the model, we normalize these values:  [[0,1],[0,1]] for bboxes and [0,1] for timestamps. We then use the parser to extract these values from the generated responses, and the verifier compares them to the ground truth using metrics such as Intersection over Union (IoU), with a threshold (e.g., IoU > 0.5) to determine correctness.
> > >
> > > Label Rationalization tends to produce very high IoU values, as the model can directly incorporate the provided labels. Answer Generation initially shows lower accuracy but improves across cycles. For example, the IoU for bounding boxes increases significantly from Cycle 1 to Cycle 2. In the following, we visualize the BBOX IoU as histograms across cycles:
> > >
> > >
> > > 0 cycle: no answer generation, only label rationalization works.
> > > 1 cycle:
> > >
> > > ```
> > > 0.5 | ▓▓▓▓▓▓▓
> > > 0.6 | ▓▓▓▓▓▓▓
> > > 0.7 | ▓▓▓▓▓▓▓
> > > 0.8 | ▓▓▓▓▓▓▓
> > > 0.9 | ▓▓▓▓▓▓▓▓▓
> > > 1.0 | ▓▓▓▓▓▓▓▓▓▓
> > >      ---------------
> > >      0  25  50  75
> > >         Frequency
> > > ```
> > >
> > >
> > > 2 cycle:
> > >
> > > ```
> > > 0.5 | ▓▓▓▓
> > > 0.6 | ▓▓▓▓
> > > 0.7 | ▓▓▓▓▓
> > > 0.8 | ▓▓▓▓▓▓
> > > 0.9 | ▓▓▓▓▓▓▓▓▓▓▓▓
> > > 1.0 | ▓▓▓▓▓▓▓▓▓▓▓▓
> > >      ---------------
> > >      0  50  100  150
> > >         Frequency
> > > ```
> > >
> > >
> > > As can be seen, the bbox IoU grows over cycles. In the first cycle, no bounding boxes were correctly predicted. In the second cycle, an almost uniform distribution can be seen. In the 2nd cycle, the abs number of correctly predicted bbox grows significantly, a relatively large spike around 1 can be seen.
> > >
> > >
> > > Once again, we thank you for your time and valuable feedback. We believe that our revisions have significantly strengthened the paper, and we hope it meets with your expectations.

---

> > > > ### Comment · Reviewer_ke7n · 2024-11-21
> > > > **Concerns addressed**
> > > >
> > > > I thank the authors for their response. My concerns are resolved now.

---

> > > > > ### Author Response · Authors · 2024-11-22
> > > > > **Thank you for your response**
> > > > >
> > > > > Dear Reviewer ke7n,
> > > > >
> > > > > We thank you for your thorough and constructive review of our submission. Your insightful feedback has improved our study, helping us enhance our baseline comparisons, add critical ablations to our research, and conduct other important analyses of Video-STaR.
> > > > >
> > > > > Thank you again for your valuable insights, time, and consideration,
> > > > >
> > > > > The Authors

---

### Official Review · Reviewer_gCco · 2024-11-03

**Soundness:** 2
**Presentation:** 3
**Contribution:** 3
**Rating:** 6
**Confidence:** 3

**Summary:**

This paper addresses the challenge of leveraging existing video datasets with diverse non-instruction labels to enhance video instruction tuning. The authors introduce Video-STaR, an automated loop for instruction generation, filtering, and self-tuning. Notably, they incorporate a Label Rationalization step within the loop to backtrace the rationale from video labels on challenging data. Experiments demonstrate that Video-STaR enables Large Multimodal Models to adapt to new tasks effectively.

**Strengths:**

1. This paper addresses a significant issue: how to utilize existing video dataset labels to improve instruction tuning. This paper presents the first self-training pipeline that could inspire future approaches in this area
2. The writing is clear. The authors also conduct extensive experiments to validate the proposed Video-STaR approach.

**Weaknesses:**

1. **Missing Experiments**: There is a lack of validation on the effectiveness of label filtering, including an ablation study on the verifier and clarification on why ground-truth (GT) labels should be included in the answers.
2. **Heuristic and Inflexible Label Verification**: The label verification process is heuristic, limiting flexibility and scalability. Since different videos contain diverse labels, the proposed approach requires custom parsers and verifiers for varying label formats. For example, timestamps may need distinct regex patterns due to format variations. Additionally, label matching, such as “chopping wood” vs. “smashing,” as seen in Table 2, appears to demand considerable human effort.
3. **Weak Baseline in Video-LLaVA Gemini**: Video-LLaVA Gemini is a relatively weak baseline. In Table 4, the authors used Gemini-pro to label only 2K samples for fine-tuning, a notably smaller dataset than those used for other baselines. The choice of 2K samples is not straightforward. The authors should attempt to label more videos with Gemini and include a curve showing the relationship between labeled data quantity and fine-tuning performance. How much labeled data is needed to achieve results comparable to Video-STaR’s 550K in both Table 3 and Table 4?
4. **Marginal Cross-Dataset Generalization (Table 4 & Table 7)**: Video-STaR’s improvement in cross-dataset generalization appears marginal. The authors attribute this to noise within the benchmark itself. If this is the case, comparisons and ablation studies across these four datasets may not convey valuable information. The authors could consider using alternative high-quality datasets to better evaluate Video-STaR’s true effectiveness in enhancing cross-dataset generalization.
5. **Typos**: Table index is incorrect. Line 260 Table 2.3 --> 2. Line 306 Table 3.2 --> 2

**Questions:**

1. Recent studies indicate that using data generated by large language models (LLMs) for self-training can ultimately have a catastrophic impact on model performance. How do the authors address this issue? An interesting experiment could involve alternately training two models—Video-LLaVA (Model A) and another model (Model B)—using the Video-STaR strategy. In this setup, results generated by Model A could fine-tune Model B, and subsequently, results from Model B could fine-tune Model A. Observing the outcomes could reveal any differences in performance and stability under this iterative training approach.

---

> ### Author Response · Authors · 2024-11-20
> **[1/3] Response to  Reviewer gCco**
>
> We sincerely thank the Reviewer gCco for their thoughtful and constructive feedback. We appreciate your recognition of our work addressing the significant challenge of leveraging existing video datasets to improve instruction tuning. We are glad that you found our writing clear and our experiments extensive. In response to your valuable feedback, **we have conducted multiple additional experiments, including training two new models, and have updated the manuscript accordingly**.
>
> ### 1. Missing Experiments: Validation of Label Filtering and Inclusion of Ground Truth Labels
> We appreciate your suggestion to provide validation on the effectiveness of our label filtering process and the rationale for including ground truth (GT) labels in the answers. In response, we conducted additional experiments to validate the parser-verifier and to analyze the correlation between the presence of GT labels in the answers and the correctness of those answers. **These results have been added to the revised manuscript.**
>
> **(i) Parser-Verifier Validation:**
> We randomly selected 500 videos per dataset to validate the parser and prompted GPT-4o to verify whether the label appears in the response (True Positives). To test for False Positives, we selected 500 videos per dataset that the parser filtered out and again used GPT-4o to check for any incorrectly filtered instances.
>
> Table: Verifying the function of the Parser-Verifier. (left) the percentage of filtered answers (answers that were categorized to have had the gold labels in them) that had the ground-truth labels. (right) percentage of filtered-out data (answers that the parser-verifier determined the labels were not in the generated answer) that had the gold labels.
>
> |       | Correctly | identified | label | Label  | not detected | |
> |:-----------|:----------|:-------------------|:-----------------|:----------------|:------------------|:--------------------|
> |    Cycle   | STARB   | Kinetics              | FineDiving       | STARB             | Kinetics               | FineDiving       |
> | 0     | 96.7                  | 99.5                  | 98.0                 | 4.8                       | 0.3                       | 1.2                       |
> | 1     | 97.4                  | 98.2                  | 96.4                  | 7.7                       | 2.2                        | 0.7                       |
> | 2     | 99.3                 | 99.4                  | 99.3                 | 3.6                       | 1.0                        | 1.2                      |
> * Filtered Dataset (True Positives): Across all datasets (Kinetics700, STAR-Benchmark, FineDiving), the parser correctly identified the presence of the label in over 97% of the cases across all cycles, indicating high precision.
> * Filtered-Out Dataset (False Positives): Only <8% of the filtered-out answers had the correct GT labels. STARB has multiple labels/answers, which is why it had the highest rate of misidentified labels.
>
> These results validate the effectiveness of our parser-verifier in accurately identifying the inclusion of GT labels in the generated answers.
>
> **(ii) Correlation Between GT Labels and Correctness:**
> We investigated whether including GT labels in the answers correlates with higher correctness. We randomly selected 500 videos per dataset and categorized the answers based on whether they included the GT label. We then prompted GPT-4o to assess the correctness of these answers.
>
> Table: Validating that label presence is correlated with the correctness of the answer. (left) The correctness rate is when the label is present in the generated answer. (right) correctness rate when the label is not present in the generated answer. In both Kinetics and FineDiving, labels are closely related to the answer, so we have a low correctness rate for answers without labels. In STAR, they are less related, so there is a slightly lower correctness rate even for answers containing the GT labels.
>
>
> |       | Correct  & | Label        | Present | \| |    Correct & | Label Not |  Present   |
> |-------|---------|----------|--------------|-|--------|----------|-----------------|
> |    Cycle   | STARB   | Kinetics | FineDiving  |\|  | STARB   | Kinetics | FineDiving       |
> | 0     | 93.2    | 98.7     | 95.9       | \|  | 8.5    | 1.8      | 0.9              |
> | 1     | 91.8    | 97.3     | 94.4       | \|  | 9.2     | 0.7      | 1.4              |
> | 2     | 92.6    | 96.5     | 96.9        | \| | 10.9   | 2.5      | 0.8              |
>
>
> * **Answers Containing GT Labels:** Over 95% of these answers were correct across all datasets, indicating a strong correlation between the presence of GT labels and answer correctness.
> * **Answers Not Containing GT Labels:** Only about 10% of these answers were correct, suggesting that the absence of GT labels in the answer often leads to incorrect responses. We believe that the low percentages of Kinetics and FineDiving are due to how similar the answers and labels are.

---

> > ### Author Response · Authors · 2024-11-20
> > **[2/3] Response to  Reviewer gCco**
> >
> > ### 2. Heuristic and Inflexible Label Verification
> >
> > The design for different label formats does require a little human effort, but it can work automatically after setting up. In Tabel 2, the parser-verifier employs word embeddings to verify the answers rather than relying on exact string matching or human efforts. That’s why the verifier still classified the example as correct (even though the exact label did not appear in the response, e.g., “chopping wood” vs. “smashing”).
> >
> > Further Automation with Small LLMs:
> > In order to try and alleviate such concerns further, we have tested the use of small Large Language Models (LLMs), such as Qwen2.5-1.5B-instruct, as a full-on replacement of the parser verifier. Qualitatively, we find that they function decently for most label types, apart from more ‘structured’ label types such as bounding boxes. We will include this option in our code release to facilitate easier adoption and adaptation by future researchers.
> >
> >
> > ### 3. Weak Baseline in Video-LLaVA Gemini
> >
> > We appreciate your feedback regarding the baseline comparison with Video-LLaVA^Gemini. To address this concern,
> > **we improved the Gemini distillation baseline (Video-LLaVA^Gemini), and updated the manuscript accordingly.**
> >
> > Improved Video-LLaVA^Gemini Baseline:
> > * **Model Initialization:** Instead of starting from the Video-LLaVA pretrained model, we initialized from the SFT variant, which has already seen approximately 100K videos. This provides a stronger baseline.
> > * **Increased Labeled Data:** We labeled an additional 8K videos using Gemini, totaling 10K videos for fine-tuning. While labeling more videos is ideal, it is cost-prohibitive (approximately $1,000 USD for 10K videos), limiting our ability to scale this further.
> > * **Training Procedure:** We fine-tuned the model on a mixture of 50% Gemini-labeled data and 50% from the original Video-LLaVA mixture to mitigate catastrophic forgetting.
> >
> > Table:  Results of improved baseline (Gemini Distillation) on the adapted dataset
> >
> > | Methods         |     Kinetics700-QA      | | STAR-bench-QA        | | FineDiving-QA     | |
> > |---------------------|----------|-------------|----------|-------------|-----------|----------|
> > |                     | Accuracy | Score    | Accuracy | Score    | Accuracy | Score    |
> > | Video-LLaVA         | 50.0     | 3.2      | 24.9     | 2.6      | 17.1     | 2.2      |
> > | Video-LLaVA+        | 49.5     | 3.2      | 28.8     | 2.8      | 17.6     | 2.2      |
> > | *Vid-LLaVA^Gemini* | *49.4*     | *2.7*      | *29.3*     | *2.6*      | *16.5*     |  *2.1*      |
> > | **Video-STaR**      | **59.9** | **3.5**  | **33.0** | **2.9**  | **21.7** | **2.3**  |
> >
> >
> > Table:  Results of improved baseline (Gemini Distillation) on TempCompass
> >
> >
> > | Methods              | Action  |       | Direction  |   | Speed  |       | Event    |  Attribute |  |  | | Avg.  |
> > |----------------------|------|----------|------|--------|------|------|--------|--------|------|--------|-------|-------|
> > |                              | Fine | Coarse  | Obj. | Cam. | Abs. | Rel. | Order | Color | Size | Both | Other |
> > | Random               | 39.7 | 40.1     | 39.8 | 39.0 | 40.8 | 42.0 | 41.5  | 40.4  | 39.9 | 38.9 | 39.4  | 40.5  |
> > | mPLUG-Owl      | 48.8 | 66.1     | 38.7 | 36.8 | 42.2 | 38.4 | 42.0  | 41.7  | 44.7 | 41.9 | 39.9  | 44.4  |
> > | Video-LLaVA    | 63.4 | 93.5     | 36.1 | 34.8 | 42.7 | 26.5 | 39.1  | 52.6  | 37.1 | 43.3 | 33.3  | 45.7  |
> > | Video-LLaVA+         | 62.1 | 93.0     | 35.0 | 32.6 | 41.1 | 38.7 | 36.4  | 59.0  | 40.2 | 36.7 | 44.4  | 47.2  |
> > | *Vid-LLaVA^Gemini*   | *65.7* | *90.5*  | *38.9* | *46.0* | *41.8* | *42.4* | *41.0* | *48.7* | *49.1* | *32.5* | *40.4* | *48.7* |
> > | **Video-STaR**       | **68.6** | **94.1** | **39.9** | **39.0** | **40.7** | **43.0** | **41.1** | **53.8** | **48.5** | **45.0** | **55.6** | **51.8** |
> > | Gemini-1.5     | 94.8 | 98.4     | 43.6 | 42.4 | 65.3 | 48.7 | 55.6  | 79.5  | 59.8 | 70.0 | 66.7  | 66.0  |

---

> > > ### Author Response · Authors · 2024-11-20
> > > **[3/3] Response to  Reviewer gCco**
> > >
> > > Table:  Results of improved baseline (Gemini Distillation) on  Zero-shot Video QA benchmarks
> > >
> > > | Methods              | Dataset Size | MSVD-QA  |         | MSRVTT-QA     |    | TGIF-QA     |     | ActivityNet-QA |   |
> > > |----------------------|--------------|----------|-------|----------|-------|----------|-------|----------|-------|
> > > |                      |              | Accuracy | Score | Accuracy | Score | Accuracy | Score | Accuracy | Score |
> > > | VideoChat     | 4K           | 56.3     | 2.8   | 45.0     | 2.5   | 34.4     | 2.3   | -        | 2.2   |
> > > | Video-LLaMA   | 4K           | 51.6     | 2.5   | 29.6     | 1.8   | -        | -     | 12.4     | 1.1   |
> > > | Video-ChatGPT  | 100K         | 64.9     | 3.3   | 49.3     | 2.8   | 51.4     | 3.0   | 35.2     | 2.7   |
> > > | Video-LLaVA*   | 100K         | 69.7     | 3.9   | 57.4     | 3.5   | 46.5     | 3.3   | 43.2     | 3.4   |
> > > | Video-LLaVA+         | 650K         | 67.8     | 3.8   | 56.0     | 3.4   | 46.5     | 3.3   | 42.2     | 3.3   |
> > > | *Vid-LLaVA^Gemini* | *128K*  | *67.2*     | *3.9*   | *56.9*     | *3.4*   | *45.5*     | *3.4*   | *42.6*     | *3.3*   |
> > > | **Video-STaR**       | **550K**     | **71.3** | **4.0** | **58.2** | **3.5** | **47.3** | **3.3** | **43.2** | **3.3** |
> > > | Gemini-1.5-pro | -            | 71.6     | 3.9   | 52.6     | 3.2   | 45.0     | 3.1   | 56.7     | -     |
> > >
> > >
> > >
> > > Overall, the Vid-LLaVA^Gemini baseline aims to mirror LIMA-style (https://arxiv.org/abs/2305.11206) distillation more closely rather than ultimately generate a large dataset using Gemini. We are confident that, at the right scale, this ultimately will outperform Video-STaR. However, this approach is very expensive, requiring thousands of dollars in API costs, and cannot be used to improve frontier models, such as Gemini itself.
> > >
> > > ### 4. Marginal Cross-Dataset Generalization
> > >
> > > Our method shows a notable improvement on TempCompass, reinforcing its effectiveness in enhancing cross-dataset generalization. In terms of Tables 4 and 7, we follow previous work (https://arxiv.org/abs/2311.10122) and use these datasets, but more recent work has found that these are saturated (https://arxiv.org/abs/2405.07798), which is why smaller performance gains are seen.
> > > We included the TempCompass dataset in our experiments to provide a more comprehensive evaluation of our method’s effectiveness. TempCompass is a high-quality benchmark designed to assess comprehensive video understanding and reasoning, including Action, Direction, Speed, Event, and Attribute Change. In this dataset, Video-STaR showed a 6.1% improvement.
> > >
> > >
> > > Table:  Results of improved baseline (Gemini Distillation) on TempCompass
> > >
> > >
> > > | Methods              | Action  |       | Direction  |   | Speed  |       | Event    |  Attribute |  |  | | Avg.  |
> > > |----------------------|------|----------|------|--------|------|------|--------|--------|------|--------|-------|-------|
> > > |                              | Fine | Coarse  | Obj. | Cam. | Abs. | Rel. | Order | Color | Size | Both | Other |
> > > | Random               | 39.7 | 40.1     | 39.8 | 39.0 | 40.8 | 42.0 | 41.5  | 40.4  | 39.9 | 38.9 | 39.4  | 40.5  |
> > > | mPLUG-Owl      | 48.8 | 66.1     | 38.7 | 36.8 | 42.2 | 38.4 | 42.0  | 41.7  | 44.7 | 41.9 | 39.9  | 44.4  |
> > > | Video-LLaVA    | 63.4 | 93.5     | 36.1 | 34.8 | 42.7 | 26.5 | 39.1  | 52.6  | 37.1 | 43.3 | 33.3  | 45.7  |
> > > | Video-LLaVA+         | 62.1 | 93.0     | 35.0 | 32.6 | 41.1 | 38.7 | 36.4  | 59.0  | 40.2 | 36.7 | 44.4  | 47.2  |
> > > | *Vid-LLaVA^Gemini*   | *65.7* | *90.5*  | *38.9* | *46.0* | *41.8* | *42.4* | *41.0* | *48.7* | *49.1* | *32.5* | *40.4* | *48.7* |
> > > | **Video-STaR**       | **68.6** | **94.1** | **39.9** | **39.0** | **40.7** | **43.0** | **41.1** | **53.8** | **48.5** | **45.0** | **55.6** | **51.8** |
> > > | Gemini-1.5     | 94.8 | 98.4     | 43.6 | 42.4 | 65.3 | 48.7 | 55.6  | 79.5  | 59.8 | 70.0 | 66.7  | 66.0  |
> > >
> > > ### 5. Wrong index.
> > > Thank you. We will fix this in the revised manuscript.
> > >
> > > ## Questions:
> > > Model collapse in self-training.
> > >
> > > Thanks for pointing out the potential negative impacts of using LLM-generated data for self-training, this actually demonstrates the urgency of our design with self-training guided by human-generated labels. In contrast to only depending on self-generated data, Video-STaR leverages existing video labels and metadata to generate instruction data. The inclusion of ground truth information anchors the generated data, mitigating model collapse.
> > >
> > >
> > > **Regarding the Suggested Experiment:**
> > >
> > > While the proposed experiment of alternately training two models using the Video-STaR strategy is intriguing, we believe that the risk of model collapse is mitigated in our approach due to the reasons mentioned above. Nonetheless, exploring such an experiment could provide additional insights, and we consider it a valuable direction for future work.
> > >
> > >
> > > Your constructive feedback has been invaluable, and we hope the improved manuscript aligns with your expectations.

---

> > > > ### Author Response · Authors · 2024-11-22
> > > > **Following up - additional comments or concerns?**
> > > >
> > > > Dear Reviewer gCco,
> > > >
> > > > Thank you for taking the time to review our work and providing valuable feedback. In response to your comments, we have conducted additional experiments focusing on validating, analyzing  and clarifying Video-STaR, analyzing the generations from Answer Generation and Label Rationalization, and strengthening our baseline comparisons. These updates have been incorporated into the revised manuscript to address your concerns.
> > > >
> > > > We hope that these revisions satisfactorily resolve the issues you raised. If there are any remaining concerns, please let us know, and we will be happy to make further improvements, including running additional analyses and experiments.
> > > >
> > > > Thank you again for your valuable insights,
> > > >
> > > > The Authors

---

> ### Comment · Reviewer_gCco · 2024-11-27
>
> The author's reply addressed some of my concerns. So I decided to increase my rating. However, considering that the method proposed in this paper still requires dedicated human efforts, it is not a fully automated workflow. It still has obvious limitations that need to be noted.

---

> > ### Author Response · Authors · 2024-11-27
> > **Responce to Reviewer gCco - thank you and follow up**
> >
> > Dear Reviewer gCco,
> >
> > Thank you for your feedback and for updating your rating. We truly appreciate your thoughtful insights and constructive comments.
> >
> > Our parser-verifier only requires human effort in the initial setup phase when new annotation types are defined, but it is fully automated after that for any dataset using the same labels. To reduce this reliance, we have explored using small LLMs, such as Qwen2.5-1.5B-instruct, as replacements for the parser-verifier and found that it can eliminate any human effort to make Video-Star a fully automatic framework. We will add a discussion about this in the Appendix.
> >
> > Please let us know what your other concerns are. We would like to do our best to ensure that you feel your concerns have been adequately addressed. Your feedback is invaluable, and we are committed to continuously improving our work.
> >
> > Thank you once again for your time, support, and constructive feedback.
> >
> > The Authors

---

### Official Review · Reviewer_AVhL · 2024-11-04

**Soundness:** 3
**Presentation:** 2
**Contribution:** 2
**Rating:** 5
**Confidence:** 4

**Summary:**

Large Multi-modal Models have achieved great success recently. However, when the problem comes to improve the reasoning capabilities of LMMs, the first and huge challenge is to get large scale and high quality datasets. Especially for video modalities, the problem is more hard by the difficulty and time-consuming. In this paper, the authors proposed Video-STaR, which is a method to generate video datasets with rationales based on LMMs. Besides, the authors releases a new video datasets named VSTAR-1M generated by the method.

**Strengths:**

* The topic of automatically generating video datasets with accurate rationales is interesting and important for future multi-modality researches. It is very difficult to use human labor to fine-grain annotate videos with various reasoning details.
* The authors release a large scale video instruction tuning dataset with rich CoT reasoning which significant saves the time and computing resources for future works.

**Weaknesses:**

* Limited technical novelty. The idea of this paper is very similar to STaR and all the proposed modules including answer generation, label rationalization and label verification are processed only for text modality, with no special considerations for video modality. It means that it is easy to transfer from normal STaR. Besides, for the methods section, the authors description is not clear and it is better to write it with more details
* Lack of experimental description, theoretical analysis, or other deeply analysis of some assumptions. In the paper, the authors claim that Label Rationalizations have higher possibility to result in hallucination. However, the authors don't provide further analysis, nor do some experiments to prove it (even only on some partial samples and do significance testing). Rather, the authors propose a case to show the problem in Figure.3. However, in cycle 1 I think the problem lays in LMMs repeats the questions instead of hallucination, and in cylce 2, Answer Generation also get a non-perfect result "with an overall score of 64.68" as the ground truth is 65.6

**Questions:**

* The VSTAR-1M has about 1M videos mentioned in the paper but the experiments only tuned with 550k videos shown in Table 4. May I ask the authors if we apply the full VSTAR videos to tune LMMs, whether the results would be better?
* Since neither the paper nor the appendix describes the verify process carefully, I briefly skimmed through the code and found that there is a 'verifier_type' for each sample. Is the type manually labeled for different samples or is it automatically judged by some rules?
* In the paper it is mentioned that bounding boxes can be used as labeled data, so does this data enable LMMs to localize objects better than before? For example, to get better results when doing video segmentation or object tracing or simply image object detection?

---

> ### Author Response · Authors · 2024-11-20
> **[1/3] Response to  Reviewer AVhL**
>
> We sincerely thank the Reviewer AVhL for their thoughtful and constructive review of our work. We appreciate the reviewer's recognition of the importance of our topic and the strengths they have highlighted. In response to their valuable feedback, we have conducted additional experiments and updated the manuscript accordingly. The reviewer's insights have significantly helped us improve the clarity and depth of our paper, and we have addressed their concerns and questions in detail below.
>
> ### 1. (a) Technical novelty
>
> To the best of our knowledge, our work is the first to explore the potential for self-improvement in LMMs, specifically focusing on the video modality and emphasizing the self-training of video understanding capabilities. While our approach draws inspiration from STaR, we believe that our method presents significant advancements specific to multi-modality. Unlike text-based data, video datasets often contain labels or metadata but lack question-answer (QA) pairs. This fundamental difference required us to redefine the problem and develop new techniques tailored to videos.
> Some differences we had to introduce to adapt to videos are:
>
> * Multimodal vs Text-only. While STaR is applied only to text-only LLMs, Video-STaR is applied to Large Multi-modal models, specifically video (and easily translatable to image-LMMs).
> * Answers vs Labels. While STaR is provided directly with QA pairs, we only had video labels to guide generation. As such, we have shown that answer generations containing labels result in more correct answers, not only CoT (see Weakness 2, response (ii), and the new experiment testing answer correctness).
> * Addressing video-specific challenges. We designed the temporal reasoning and perception Q-A generation for videos, including object comparisons, where objects at different times and locations are compared, and grounded captioning. Please refer to the revised manuscript Table 8 for a complete list of all the design question types.
> * Developed a parser-verifier specific to video data: STaR checks if the final answer is correct. However, most video datasets do not have QA pairs. Instead, we must first identify the predicted labels in the generated text. For this, we trained many NER models or used regex/BERT embeddings to identify the labels and then compare them correctly to the gold labels.
> * Video-Specific, down-stream applications: We demonstrated the utility of Video-STaR to adapt LMM to downstream applications, like acting like a judge in Olympic diving events.
>
>
> These adaptations aim to tackle the different video-specific challenges. In video, there are many challenges related to generating conversations related to video content. Self-training could alleviate these challenges, particularly when leveraging readily available (or generable) supervision.
>
>
> ### 1. (b) The method is unclear
>
> We added a detailed description of the Parser-Verifier to the appendix (please see App. Sec. F). If any other part of our work needs clarification, we will happily update those sections.

---

> ### Author Response · Authors · 2024-11-20
> **[2/3] Response to  Reviewer AVhL**
>
> ### 2. (a) Lack of experimental description, theoretical analysis, or other deep analysis of some assumptions.
>
> We appreciate your suggestion to provide more detailed experimental descriptions and analyses. In response, **we have conducted additional experiments and analyses to validate our claims and provide deeper insights into our assumptions. These will be included in the revised manuscript**.
>
>
> **(i) Higher Possibility of Hallucination in Label Rationalizations:**
>
>
> We conducted a new experiment to test hallucinations and demonstrate that this is mostly an issue during Label Rationalization, while the hallucination rate in Answer Generation is low. **This experiment is included in the revised manuscript**. While a ~10% hallucination rate in Label Rationalization may appear to be problematic, Label Rationalization is only used to bootstrap Answer Generation. The final dataset does not include any Label Rationalization-based generations, so hallucinations will not impact the final model performance.
>
> We randomly selected *500 instances of Answer Generations and 500 instances of Label Rationalizations per cycle and per dataset* (Kinetics700, STAR-Benchmark, and FineDiving). We utilized GPT-4o as an evaluator by providing it with the question, the model’s answer, and the sampled video frames (ensuring alignment with our model’s inputs). We asked GPT-4o to identify any textual descriptions in the answers that did not appear in the video.  The results of these experiments can be seen here:
>
> Table: Hallucination rate in Answer Generation and Label Rationalization Across Cycles. For kinetics, we did not use label rationalization. This experiment is included in our revised manuscript.
>
> | cycle | | **Answer** | **Generation**  | \| |  **Label** | **Rationalization** |
> |-------|-------------------|---|---|---------------------------|---|---|
> |          | STAR      | Kinetics | FineDiving | \| | STAR      | FineDiving |
> | 0       | 1.25        | 1.0         | 0.5            | \| | 11.4        | 10.2          |
> | 1       | 0.5          | 2.25       | 3               | \| | 6.2          | 14.8          |
> | 2       | 1.75         | 0.5         | 1.5            | \| | 14.1        | 7.1            |
>
>
> The higher hallucination rate in Label Rationalizations arises because the model may rely solely on the provided labels and neglect the video content. In contrast, during Answer Generation, the model must reference the video to produce coherent answers, reducing the likelihood of hallucinations. As such, almost no (~1%) hallucination rate can be seen in Answer Generation, while in Label Rationalization, ~10% hallucination rate can be observed. In Kinetics, we did not use Label Rationalization as direct Answer Generation had a high initial conversion rate.
>
> **(ii) Generations Containing Ground Truth Labels Have Higher Correctness Probability**
>
> Another core assumption is that answers that contain ground-truth labels are more likely to be correct. To show this is the case, we conducted the following experiment. **These results are included in the revised manuscript**. We randomly selected 500 answers that contain and do not contain the gold labels, per dataset and cycle. We then used GPT-4o to assess the correctness of these answers :
>
>
> Table: Validating that label presence is correlated with the correctness of the answer. (left) The correctness rate is when the label is present in the generated answer. (right) correctness rate when the label is not present in the generated answer. In both Kinetics and FineDiving, labels are closely related to the answer, which is why we have such a low correctness rate for answers without labels. In STAR, they are less related, so there is a slightly lower correctness rate even for answers containing the GT labels.
>
>
> |       | Correct  & | Label        | Present | \| |    Correct & | Label Not |  Present   |
> |-------|---------|----------|--------------|-|--------|----------|------------------|
> |    Cycle   | STARB   | Kinetics | FineDiving  |\|  | STARB   | Kinetics | FineDiving       |
> | 0     | 93.2    | 98.7     | 95.9       | \|  | 8.5    | 1.8      | 0.9              |
> | 1     | 91.8    | 97.3     | 94.4       | \|  | 9.2     | 0.7      | 1.4              |
> | 2     | 92.6    | 96.5     | 96.9        | \| | 10.9   | 2.5      | 0.8              |
>
>
> * Answers Containing GT Labels: Over 95% of these answers were correct across all datasets, indicating a strong correlation between the presence of GT labels and answer correctness.
> * Answers Not Containing GT Labels: Only about 10% of these answers were correct, suggesting that the absence of GT labels in the answer often leads to incorrect responses. We believe that the low percentages of Kinetics and FineDiving are due to how similar the answers and labels are.
>
>
>
> If Reviewer AVhL would like us to run additional experiments testing other aspects of our work, we are open to doing so.

---

> > ### Author Response · Authors · 2024-11-20
> > **[3/3] Response to  Reviewer AVhL**
> >
> > ### 2. (b)  "However, in cycle 1 I think the problem lays in LMMs repeats the questions instead of hallucination, and in cycle 2, Answer Generation also get a non-perfect result "with an overall score of 64.68" as the ground truth is 65.6”
> >
> > While the response is non-perfect, it is extremely difficult to estimate the score (range from 0 to 100) of an Olympic diving event from the video itself. The difference between 64.68 and 65.6 is just 1% compared to the range. This, on its own, is non-trivial, so we did not require the models to perfectly predict final scores (other labels are treated in this manner, e.g., with bounding boxes, we used an IoU threshold of 0.5).
> >
> > ## Questions
> > ### 1. Application of the Full VSTAR Dataset.
> > We would like to clarify that the full VSTAR dataset was used in our experiments. The VSTAR-1M dataset consists of approximately **1 million instruction tuning pairs** derived from about 550,000 unique videos. The total number of videos in our source datasets is around 675,000 (as shown in Table 1 of the manuscript). After two self-training cycles, we generated 1 million instruction pairs using 550,000 videos, which we used for training.
> >
> > ### 2. Verification Process Details
> >
> >
> > Thank you for bringing to our attention the need for more clarity regarding the verification process. In our approach, the `verifier_type` for each sample is manually labeled. We identified several label types relevant to our datasets: bounding boxes, temporal action localization, action sequence, object comparison, caption, timestamped caption, action quality assessment, action after/before action, and action classification.
> >
> >
> > Each sample is assigned a `verifier_type` based on these categories, which guides the verification process. We have updated the appendix to include a detailed description of the verification process, including the criteria for each `verifier_type` and how it is applied (see Sec. F).
> >
> > The effort of setting up the parser-verifier for new label types is marginal compared to manually labeling VQA data. In order to try and alleviate such concerns further, we have tested the use of small Large Language Models (LLMs), such as Qwen2.5-1.5B-instruct, as a full-on replacement of the parser verifier. Qualitatively, we find that they function decently for most label types, apart from more ‘structured’ label types such as bounding boxes. We will include this option in our code release to facilitate easier adoption and adaptation by future researchers.
> >
> > ### 3. Impact on Object Localization and Grounding
> >
> > While our dataset includes some bounding box data, our primary focus was improving instruction tuning rather than enhancing model grounding capabilities. Consequently, the amount of data related to object localization is relatively small. As such, we did not observe significant improvements in tasks like video segmentation, object tracking, or image object detection.
> > However, we recognize the potential of our approach to contribute to these areas. We are working on extensions to Video-STaR that focus specifically on temporal grounding and object localization tasks. By incorporating more extensive and diverse bounding box annotations and related data, we aim to enable LMMs to better localize objects and understand spatial-temporal relationships in videos.
> >
> >
> > We thank Reviewer AVhL for their valuable insights and believe our work is stronger after adding the additional experiments.

---

> > > ### Author Response · Authors · 2024-11-22
> > > **Following up - additional comments or concerns?**
> > >
> > > Dear Reviewer AVhL,
> > >
> > > Thank you for taking the time to review our work and providing valuable feedback. In response to your comments, we have conducted additional experiments focusing on enhancing our experimental descriptions and providing deeper theoretical analyses of our assumptions. These updates have been incorporated into the revised manuscript.
> > >
> > > We hope that these revisions satisfactorily resolve the concerns you raised. If there are any remaining concerns, please let us know, and we will be happy to make further improvements, including running additional analyses and experiments.
> > >
> > > Thank you again for your valuable insights,
> > >
> > > The Authors

---

> > > > ### Comment · Reviewer_AVhL · 2024-11-27
> > > > **Thank the author for the response**
> > > >
> > > > Thank the authors for the detailed response and addressed some of my concerns. However, the contribution still lies mostly in the text modality. Additionally, weakness 2 is not fully addressed. I think this is a borderline paper and maintain my original score.

---

> ### Author Response · Authors · 2024-11-27
> **Thank you for your response**
>
> Dear Reviewer AVhL,
>
> Thank you for your response and feedback. We want to emphasize the contribution of this work. Collecting video instruction tuning datasets is incredibly time-consuming, requiring annotators to view videos and generate question-answer (QA) pairs manually. Unlike images, which can be reviewed in a few seconds, videos demand significantly more time and effort. Therefore, we believe self-training is a promising approach to introducing human supervision to the video-SFT dataset. Currently, no self-training methods are available for video-LMMs, and Video-STaR represents the first exploration in this research direction. Video-STaR can be further scaled up to internet-scale datasets using any available metadata (e.g., YouTube transcriptions, video text) or using auxiliary models (e.g., OWLv2 to generate bbox, OCR, etc).
>
> **Regarding point 2**, where you mentioned a "lack of experimental description, theoretical analysis, or other deep analysis of some assumptions,". As a result, we tested our main assumptions:
>
> 1. **Labels in Answer Generation Lead to Higher Correctness Probability**: In our rebuttal experiments, we found that answers containing the correct label have a correctness rate exceeding 90%.
> 2. **Answer Generation Has Fewer Hallucinations Than Label Rationalization**: Our experiments showed that Answer Generation had a negligible hallucination rate (<3%).
> 3. **Parser-Verifier Clarification and Validation**: The revised manuscript includes detailed descriptions, new figures, tables, and experiments validating the parser-verifier.
>
> Could you please let us know what aspects of our work we need to provide more information about experimental description, theoretical analysis, or other deep analysis of some assumptions? In the remaining rebuttal time, we would like the opportunity to improve our work further.
>
> Thank you once again for your valuable input.
>
> Best regards,
>
> The Authors

---

> ### Author Response · Authors · 2024-12-03
> **Follow-up**
>
> Dear Reviewer AVhL,
>
> Thank you once again for your thoughtful feedback and for taking the time to review our work. As today is the final day for discussion, please let us know if you have any additional questions or require further clarification. We are eager to address any remaining concerns to improve our manuscript.
>
> Thank you for your time and consideration.
>
> Best regards,
> The Authors

---

### Official Review · Reviewer_MNYz · 2024-11-07

**Soundness:** 2
**Presentation:** 2
**Contribution:** 2
**Rating:** 6
**Confidence:** 4

**Summary:**

The paper introduces a novel self-training approach, Video Self-Training with augmented Reasoning (Video-STaR), to improve the performance of Large Multi-modal Models (LMMs) in video instruction tuning. By leveraging any labeled video dataset, Video-STaR enhances video question answering and adapts LMMs to new downstream tasks. The method cycles between generating answers, verifying them using the video labels, and fine-tuning the model. Experimental results show significant improvements in video understanding tasks, such as a 6.1% increase in Video QA performance and enhanced action quality assessment accuracy.

**Strengths:**

- **Originality**: Video-STaR introduces an innovative self-training approach for video instruction tuning, enabling the use of any labeled video dataset for instruction tuning. This represents a novel solution to a critical problem in video QA and reasoning by removing the dependency on manual dataset collection.

- **Quality**: The paper is well-structured, with detailed experimental evaluations demonstrating significant improvements in video QA accuracy and adaptability to downstream tasks. It also addresses weaknesses in previous methods, such as reducing hallucinations in model answers.

- **Clarity**: The methodology, including the self-training cycles of generation, filtering, and tuning, is clearly articulated, with effective visual aids to illustrate the process.

- **Significance**: The paper contributes meaningfully to the field of video QA and LMM adaptation, showing improvements in both general video understanding and task-specific performance. It also creates a large and diverse dataset (VSTAR-1M), further enriching the resources available for multimodal research.

**Weaknesses:**

- **Computational Intensity**: The iterative nature of the self-training cycles, involving both answer generation and label rationalization, introduces significant computational overhead. This could limit the scalability and efficiency of the proposed method, especially in resource-constrained environments.

- **Hallucination in Rationalization**: The reliance on label rationalization, especially for difficult tasks, may increase the likelihood of hallucinations. This undermines the robustness of the system in generating accurate explanations and answers, particularly in complex video tasks like FineDiving.

- **Generalizability of Label Rationalization**: While the method assumes that all video labels require rationalization, certain straightforward labels may not benefit from this additional step. This leads to unnecessary computational load without proportional gains in performance.

- **Limited Dataset Variety**: The study largely focuses on specific datasets like FineDiving, STAR-benchmark, and Kinetics700. Expanding the evaluation to a broader set of tasks could help demonstrate the full generalizability and adaptability of the approach to more diverse real-world video datasets.

**Questions:**

1. **Clarification on Dataset Labeling**: Can the authors clarify how they handle datasets where video labels are ambiguous or not well-defined? It would be helpful to understand how such cases are managed during the label verification step to avoid generating incorrect instructions.

2. **Self-Training Cycles**: Could the authors provide more details on how they determine when the self-training process has plateaued? Is there a specific performance threshold or metric used to stop the cycles?

3. **Hallucination Mitigation**: The paper acknowledges hallucination as a potential issue, especially in label rationalization. Are there additional methods or heuristics that the authors could explore to reduce hallucinations during both answer generation and rationalization phases?

4. **Generalizability to Complex Tasks**: Video-STaR shows notable improvements in performance on diverse video tasks. Could the authors elaborate on how the method handles particularly complex reasoning tasks beyond action recognition and quality assessment? How might this approach generalize to more abstract tasks, such as subjective video analysis?

5. **Computational Overhead**: Given the iterative nature of the self-training process, what are the computational requirements and trade-offs for implementing Video-STaR at scale? Would this method be feasible for larger video datasets, and if so, what optimizations could be applied?

---

> ### Author Response · Authors · 2024-11-20
> **[1/2] Response to reviewer MNYz**
>
> We sincerely thank the reviewer for their thoughtful and detailed feedback. We are pleased that you found our approach original, of high quality, clear, and significant to the field. In response to your valuable comments, we have conducted additional experiments and updated the manuscript accordingly. We address your concerns and questions below.
>
>
> ### 1. Computational Intensity
>
> While our method does involve additional computation, we believe it remains practical and scalable for several reasons:
>
> (i) Compared with human-annotated instruction-tuning data and other data curation methods, Video-STaR doesn’t need human effort or expensive APIs (which also introduce implicit computational overhead) to distill frontier models.
>
> (ii) Once generated, the dataset can be re-used by other models.
>
> (iii) Practitioners can adjust the number of self-training cycles or limit the size of the datasets used based on available computational resources. This flexibility allows for practical implementation without prohibitive overhead.
>
> (iv) Recent trends in the field indicate that smaller models (e.g., less than 3 billion parameters [1,2]) are achieving competitive performance, which can reduce computational requirements. Although we did not employ techniques like Low-Rank Adaptation (LoRA) in our current implementation, such methods can be utilized in future work to decrease GPU memory usage and computational costs significantly.
>
> The self-training process in Video-STaR is inherently scalable. To adapt LMMs to new and diverse task, the scale required is easily accessible to most academics, where gathering ~100K SFT data should suffice.  However, Video-STaR could also be utilized on a much larger scale (100M>) by frontier companies to further improve their models. This scale is only accessible to a few large companies.
>
> In the revised manuscript, we discuss potential strategies (Sec. G) —such as using smaller models and parameter-efficient training methods—to mitigate computational intensity for various use cases.
>
> [1] LongVU: Spatiotemporal Adaptive Compression for Long Video-Language Understanding
> [2] Qwen2-VL: Enhancing Vision-Language Model's Perception of the World at Any Resolution
>
>
> ### 2. Hallucination in Rationalization
>
> To address Reviewer MNYz’s concern, we conducted a new experiment to test hallucinations and demonstrate that this is mostly an issue during label rationalization. **This experiment is included in the revised manuscript**. We found that hallucination rates in label rationalization are relatively low, at \~10%. However, hallucination rates in answer generation are negligible (~1%) and do not increase over cycles, even though the model is trained on label rationalizations.  While a ~10% hallucination rate in Label Rationalization may appear to be problematic, Label Rationalization is only used to bootstrap Answer Generation. The final dataset does not include any Label Rationalization-based generations, so hallucinations will not impact the final model performance.
>
> Label rationalization is only used during self-training cycles to inject complex reasoning capabilities into the model, and the answer generation is used to ensure the model would not be greatly influenced by the potential hallucination in the Label rationalization stage. This is why, in the final training stage, we exclusively use direct answer generation, ensuring that any hallucinations from rationalization do not impact the final model’s performance. We detail the experiment below:
>
> We randomly selected 500 Answer Generations and Label Rationalizations per cycle and dataset and set out to determine how many of them were hallucinated. To do this, we prompted GPT4o with the question, answer, and video. We made sure to sample the same frames our model sampled for improved alignment and asked it to identify if any textual descriptions do not appear in the video.
>
>
> Table: Hallucination rate in Answer Generation and Label Rationalization Across Cycles. For kinetics, we did not use label rationalization. **This experiment is included in our revised manuscript**.
>
> | cycle | | **Answer** | **Generation**  | \| |  **Label** | **Rationalization** |
> |-------|-------------------|---|---|---------------------------|---|---|
> |          | STAR      | Kinetics | FineDiving | \| | STAR      | FineDiving |
> | 0       | 1.25        | 1.0         | 0.5            | \| | 11.4        | 10.2          |
> | 1       | 0.5          | 2.25       | 3               | \| | 6.2          | 14.8          |
> | 2       | 1.75         | 0.5         | 1.5            | \| | 14.1        | 7.1            |
>
> ### 3. Generalizability of Label Rationalization
> We only perform label rationalization if direct answer generation has failed (refer to Fig. 2), which ensures only the question requiring additional reasoning will be rationalized.

---

> > ### Author Response · Authors · 2024-11-20
> > **[2/2] Response to reviewer MNYz**
> >
> > ### 4. Limited Dataset Variety
> >
> > We selected FineDiving, STAR-Benchmark, and Kinetics700 because they are highly representative and diverse within the video domain:
> >
> > **(i) Kinetics700:** has high video diversity and size.
> >
> > **(ii) STAR-Benchmark:** has high label diversity and temporal reasoning
> >
> > **(iii) FineDiving:** Is more application-heavy while also being very difficult, video-specific, and requires a lot of in-domain knowledge and reasoning.
> >
> > These datasets collectively demonstrate the adaptability and robustness of our approach across different types of video tasks. Nonetheless, we acknowledge the importance of evaluating our method on a broader set of datasets to further validate its generalizability. We plan to include additional diverse real-world video datasets in future work to comprehensively showcase the effectiveness of our approach.
> >
> >
> > ## Questions:
> > ### 1. Clarification on Dataset Labeling:
> >
> > The design of Video-STaR specifically handles wrong or noisy labels with a dual-verification mechanism. In Answer Generation, the model is not provided with any label when prompted, and therefore, it is unlikely that the generation is the same as wrong/noisy label, and this data point will be ultimately filtered out with our parser-verifier.
> >
> >
> > ### 2. Self-Training Cycles
> > The self-training process is done when the number of new samples generated using answer generation or label rationalization stops increasing.
> >
> > ### 3. Hallucination Mitigation:
> > To mitigate such concerns, label rationalization is only utilized during the self-training cycles to generate additional training data. In the final training stage, we exclusively use direct answer generation, ensuring that any hallucinations from rationalization do not impact the final model’s performance. As can be seen in our response to Weakness 2, there are almost no hallucinations (~1%) during answer generation. While a ~10% hallucination rate in Label Rationalization may appear to be problematic, Label Rationalization is only used to bootstrap Answer Generation. The final dataset does not include any Label Rationalization-based generations, so hallucinations will not impact the final model performance.
> >
> >
> > ### 4. Generalizability to Complex Tasks:
> >
> > Video-STaR generalizes well to complex reasoning tasks beyond action recognition. For example, both the STAR-Benchmark and FineDiving datasets require complex reasoning. STAR focuses on Situated Reasoning, offering demanding question-answering tasks, symbolic descriptions of situations, and logic-based diagnostics derived from real-world video scenarios. Meanwhile, the FineDiving dataset requires evaluating an action’s quality and understanding subtle nuances like technique, form, and execution. Our method achieved notable improvements in this complex task, demonstrating its capability to handle more abstract and subjective video analysis. Some examples of this can be seen in Figure 5 of the appendix, where the LMM needs to identify the sequence of actions, evaluate the difficulty of each one, and use this to predict a final score for the dive.
> >
> >
> > ### 5. Computational Overhead:
> > Please see the response to Weakness 1.
> >
> > Thank you for helping us enhance our work—we look forward to your favorable consideration of our revised paper.

---

> > > ### Comment · Reviewer_MNYz · 2024-11-26
> > > **Thank authors for the rebuttal.**
> > >
> > > I appreciate the authors' thorough and well-substantiated responses, which have resolved my concerns. Additionally, I have carefully reviewed the feedback from other reviewers and the detailed discussion process. While the contributions in terms of comprehensive experimental results and technical details are commendable, the evidence provided does not sufficiently justify a higher score. Therefore, I have decided to retain my score of 6.

---

> > > > ### Author Response · Authors · 2024-11-26
> > > > **Thank you for your response**
> > > >
> > > > Dear Reviewer MNYz,
> > > >
> > > > Thank you for your thoughtful feedback and the time and effort you took in reviewing our work. We appreciate that our responses have helped resolve the issues you initially raised.
> > > >
> > > > If, during the remaining time of the review process, there are any additional experiments, analyses, or clarifications regarding Video-STaR that come to mind and could help us further improve this work, we are fully commited to addressing them.
> > > >
> > > > Thank you for your time and consideration,
> > > >
> > > > The Authors

---

> ### Author Response · Authors · 2024-11-22
> **Following up - additional comments or concerns?**
>
> Dear Reviewer MNYz,
>
>
> Thank you for taking the time to review our work and for your thoughtful feedback. We have conducted additional experiments and updated the manuscript accordingly.
>
> We hope that these revisions resolve the concerns you raised. If there are any remaining questions or concerns, please let us know, and we will be happy to address them, including running additional experiments or updating the manuscript.
>
> Thank you again for your valuable insights,
>
> The Authors

---

> ### Author Response · Authors · 2024-12-03
> **Follow-up**
>
> Dear Reviewer MNYz,
>
> Thank you once again for your thoughtful feedback and for taking the time to review our work. As today is the final day for discussion, please let us know if you have any additional questions or require further clarification. We are eager to address any remaining concerns to improve our manuscript.
>
> Thank you for your time and consideration.
>
> Best regards,
> The Authors

---

### Author Response · Authors · 2024-11-20
**[1/3] General Response (Updated)**

(NOTE: for the sake of brevity, we opted to edit the overview message from Nov 20th into the final overview message for the rebuttal)

Dear Reviewers, Area Chairs, and Program Chair,

We sincerely thank you all for considering our submission and for the insightful and constructive feedback provided. We greatly appreciate the time and effort each reviewer has dedicated to evaluating our work.  The consistency of the feedback across multiple reviewers has highlighted key areas for improvement, which we have addressed in the rebuttal through additional experiments, ablations, and analyses, summarized below.

We are grateful for the reviewers' positive feedback. Video-STaR is the first video-LMM self-training method, and we were happy that the reviewers noted on Video-STaR’s contributions:


**Motivation and significance.** (Reviewers **MNYz**, **AVhL**, and **gCco**).
Reviewer **AVhL** highlighted the challenges in improving the reasoning capabilities of LMMs due to the difficulty and time-consuming nature of acquiring large-scale, high-quality video datasets.
Similarly, Reviewer **gCco** acknowledged that *"this paper addresses a significant issue: how to utilize existing video dataset labels to improve instruction tuning."*


**Importance to the field.** (Reviewers **MNYz**, **AVhL**, and **gCco**).
Reviewer **MNYz** remarked that *"the paper contributes meaningfully to the field of video QA and LMM adaptation..."*.
Reviewer **AVhL** reiterated that *"The topic of automatically generating video datasets with accurate rationales is interesting and important for future multi-modality research."*.
 Reviewer **gCco** pointed out that Video-STaR *"... could inspire future approaches in this area."*


**Novelty of Video-STaR as the first video self-training method.** (Reviewers **MNYz**, **gCco**, and **ke7n**).
Reviewer **MNYz** stated that *"Video-STaR introduces an innovative self-training approach for video instruction tuning, enabling the use of any labeled video dataset for instruction tuning. This represents a novel solution to a critical problem in video QA and reasoning by removing the dependency on manual dataset collection."*
Reviewer **gCco** observed that *"this paper presents the first self-training pipeline …"*
and Reviewer **ke7n** described our method as *"a novel method to perform instruction tuning of Large Multi-modal Models (LMMs) on labeled video datasets, where the label format is arbitrary and different from that needed for instruction tuning."*


**Implementation and experiments.** (Reviewers **gCco**, **ke7n**). Reviewer **gCco** described the experiments as ‘extensive’, while Reviewer **ke7n** expressed that *"I appreciate the ablation study."*.


**Dataset contribution.** (Reviewers **MNYz**, **AVhL**).
Reviewer **MNYz** highlighted that *"it also creates a large and diverse dataset (VSTAR-1M), further enriching the resources available for multimodal research,"* and Reviewer **AVhL** added that *"the authors release a large scale video instruction tuning dataset with rich CoT reasoning which significantly saves the time and computing resources for future works."*

**Clarity.** (Reviewers **MNYz**, **gCco**, and **ke7n**).
Reviewer **MNYz** stated that *"the methodology, including the self-training cycles of generation, filtering, and tuning, is clearly articulated, with effective visual aids to illustrate the process."* Reviewer **gCco** observed that *"the writing is clear,"* and Reviewer **ke7n** affirmed that *"the paper is well written and is easy to follow."*

In response to your valuable comments and suggestions, we have conducted several new experiments, performed additional analyses, and introduced new baselines to enhance the robustness and comprehensiveness of our submission. These enhancements have been incorporated into both the main manuscript and the appendix, resulting in a strengthened presentation of Video-STaR, and are summarized below.

---

> ### Author Response · Authors · 2024-11-20
> **[2/3] General Response (Updated)**
>
> # Experimental description, theoretical analysis, or other deep analysis of some assumptions:
>
> &nbsp;
>
> ### 1. **Hallucination Rates in Label Rationalization and Answer Generation** (Reviewers AVhL, MNYz, gCco):
>
> We conducted experiments measuring hallucination rates in both Label Rationalization and Answer Generations, and confirmed that hallucinations are primarily present during Label Rationalization (\~10% rate), whereas Answer Generation exhibits a negligible hallucination rate (~1%). Since Label Rationalization is only used to bootstrap Answer Generation and is not included in the final training data, it does not adversely affect the model's performance (see Table below or Figure 13 in the appendix).
>
>
> Table: Hallucination rate in Answer Generation and Label Rationalization Across Cycles. For Kinetics, we did not use Label Rationalization. **This experiment is included in our revised manuscript**.
>
> | cycle | | **Answer** | **Generation** | \| |  **Label** | **Rationalization** |
> |---|---|---|---|---|---|---|
> | | STAR | Kinetics | FineDiving | \| | STAR | FineDiving |
> | 0 | 1.25 | 1.0 | 0.5 | \| | 11.4 | 10.2 |
> | 1 | 0.5 | 2.25 | 3 | \| | 6.2 | 14.8 |
> | 2 | 1.75 | 0.5 | 1.5 | \| | 14.1 | 7.1 |
>
> &nbsp;
>
>
> ### 2. **Correlation Between Ground Truth Labels and Answer Correctness** (Reviewers MNYz, AVhL, gCco, ke7n)
>
> We validated our assumption that answers containing gold labels are more likely to be correct. Our experiments showed that over 95% of answers containing gold labels were correct, while only about 10% of answers without gold labels were correct. This strong correlation justifies filtering generations that include gold labels and supports the effectiveness of our parser-verifier (see Table below or figure 12 in the appendix).
>
>
> Table: Validating that label presence is correlated with the correctness of the answer. (left) The correctness rate when the label is present in the generated answer. (right) correctness rate when the label is not present in the generated answer. In both Kinetics and FineDiving, labels are closely related to the answer, which is why we have such a low correctness rate for answers without labels. In STAR, they are less related, so there is a slightly lower correctness rate even for answers containing the gold labels.
>
>
> | | Correct & | Label | Present | \| | Correct & | Label Not | Present |
> |---|---|---|---|-|---|---|---|
> | Cycle | STARB | Kinetics | FineDiving  |\|  | STARB   | Kinetics | FineDiving       |
> | 0 | 93.2 | 98.7 | 95.9 | \| | 8.5 | 1.8 | 0.9 |
> | 1 | 91.8 | 97.3 | 94.4 | \| | 9.2 | 0.7 | 1.4 |
> | 2 | 92.6 | 96.5 | 96.9 | \| | 10.9 | 2.5 | 0.8 |
>
>
> &nbsp;
>
> ### 3. **Validation of the Parser-Verifier Effectiveness  + Additional Details**  (Reviewers MNYz, gCco, AVhL, ke7n)
>
> We performed additional experiments to assess the accuracy of our parser-verifier in identifying the inclusion of GT labels in generated answers. The parser-verifier correctly identified the labels in over 97% of cases (true positives), and less than 8% of the filtered-out answers actually contained gold labels (false positives). These results confirm the high effectiveness of our parser-verifier in ensuring high-quality training data. (see Table below or Figure 14 in the appendix).
>
> To clarify how the Parser-Verifier functions, we added Sec F to the appendix, including Figure 11 and Table 8, which details all the label types identified in Video-STaR and their corresponding parser, verifier, and threshold.
>
>
> Table: Verifying the function of the Parser-Verifier. (left) the percentage of filtered answers (answers that were categorized to have had the gold labels in them) that had the ground-truth labels. (right) percentage of filtered-out data (answers that the parser-verifier determined the labels were not in the generated answer) that had the gold labels.
>
> |   | Correctly | identified | label | \|  | Label  | not detected | |
> |---|---|---|---|-|---|---|---|
> | Cycle | STARB | Kinetics  | FineDiving |  \|  | STARB | Kinetics | FineDiving |
> | 0 | 96.7 | 99.5 | 98.0 | \| | 4.8 | 0.3 | 1.2 |
> | 1 | 97.4 | 98.2 | 96.4 | \| | 7.7 | 2.2 | 0.7 |
> | 2 | 99.3 | 99.4 | 99.3 | \| | 3.6 | 1.0 | 1.2 |

---

> ### Author Response · Authors · 2024-12-04
> **[3/3] General Response**
>
> # Additional experiments
>
> &nbsp;
> ### 4. **Improved Baselines** (Reviewers gCco, ke7n)
>
> To strengthen our baseline comparisons, we improved the Video-LLaVA^Gemini baseline by initializing from the fine-tuned Video-LLaVA model, increasing the amount of Gemini-generated data, and including 50% of the original Video-LLaVA dataset (to mitigate forgetting). Despite these enhancements, our method still outperformed the improved baseline. This reinforces our approach’s practical advantages and effectiveness (see Tables bellow or Tables 3-5 in the manuscript).
>
> Table:  Results of improved Gemini Distillation baseline (Video-LLaVA^Gemini) on the adapted dataset
>
> | Methods | Kinetics700-QA  | | STAR-bench-QA | | FineDiving-QA | |
> |---|---|---|---|---|---|---|
> |  | Accuracy | Score | Accuracy | Score    | Accuracy | Score  |
> | Video-LLaVA  | 50.0 | 3.2 | 24.9 | 2.6 | 17.1 | 2.2 |
> | Video-LLaVA+  | 49.5 | 3.2 | 28.8 | 2.8 | 17.6  | 2.2 |
> | *Vid-LLaVA^Gemini* | *49.4* | *2.7* | *29.3* | *2.6* | *16.5* |  *2.1* |
> | **Video-STaR** | **59.9** | **3.5**  | **33.0** | **2.9**  | **21.7** | **2.3** |
>
>
> Table:  Results of improved Gemini Distillation baseline (Video-LLaVA^Gemini)  on TempCompass
>
> | Methods | Action | | Direction | | Speed | | Event | Attribute | | | | Avg. |
> |---|---|---|---|---|---|---|---|---|---|---|---|---|
> |  | Fine | Coarse  | Obj. | Cam. | Abs. | Rel. | Order | Color | Size | Both | Other |
> | mPLUG-Owl   | 48.8 | 66.1 | 38.7 | 36.8 | 42.2 | 38.4 | 42.0  | 41.7  | 44.7 | 41.9 | 39.9  | 44.4  |
> | Video-LLaVA   | 63.4 | 93.5 | 36.1 | 34.8 | 42.7 | 26.5 | 39.1  | 52.6  | 37.1 | 43.3 | 33.3  | 45.7  |
> | Video-LLaVA+ | 62.1 | 93.0 | 35.0 | 32.6 | 41.1 | 38.7 | 36.4  | 59.0  | 40.2 | 36.7 | 44.4  | 47.2  |
> | *Vid-LLaVA^Gemini*   | *65.7* | *90.5*  | *38.9* | *46.0* | *41.8* | *42.4* | *41.0* | *48.7* | *49.1* | *32.5* | *40.4* | *48.7* |
> | **Video-STaR** | **68.6** | **94.1** | **39.9** | **39.0** | **40.7** | **43.0** | **41.1** | **53.8** | **48.5** | **45.0** | **55.6** | **51.8** |
>
> &nbsp;
>
> ### 5. **Ablations to demonstrate the utility of each component** (Reviewers AVhL, ke7n)
>
> We conducted an additional ablation to evaluate the individual contributions of Label Rationalization and Answer Generation. Specifically, we added a model trained only with Label Rationalization, which resulted in lower performance compared to Video-STaR. This highlights the importance of both components, with Answer Generation playing a crucial role in reducing hallucinations and improving answer correctness.
>
>
> |  | MSVD|-QA | MSRVTT|-QA | TGIF|-QA | ActivityNet|-QA |
> |---|---|---|---|---|---|---|---|---|
> | | Accuracy | Score | Accuracy | Score | Accuracy | Score | Accuracy | Score |
> | **Video-STaR**   | 71.3 | 4.0 | 58.2 | 3.5 | 46.8 | 3.3 | 42.2 | 3.3 |
> | **- Generation**  | 70.6 | 3.9 | 57.8 | 3.5 | 44.9 | 3.3 | 41.1 | 3.2 |
>
>
> | | Kinetics700|-QA | STAR-|bench-QA | FineDiving|-QA |
> |---|---|---|---|---|---|---|
> | | Accuracy | Score | Accuracy | Score | Accuracy | Score |
> | **Video-STaR**  | 59.9 | 3.5 | 33.0 | 2.9 | 21.6 | 2.3 |
> | **- Generation** | 58.8 | 3.5 | 28.7 | 2.7 | 17.1 | 2.1 |
>
>
> We hope these additional experiments addressed the concerns raised and strengthen the contributions of our paper. We have updated the manuscript accordingly to reflect these improvements.
>
> Sincerely,
>
> The Authors

---

### Meta-Review · Area_Chair_qgdV · 2024-12-21

**Metareview:**

This paper proposes a self-training approach for instructing tuning MLLMs with video.

The reviewers find the approach interesting and also appreciate the dataset release.

Reviewers raised questions and concerns primarily with the experimental details and qualified baselines.  These primarily addressed by the author responses.

In the end, the paper received three positive reviews (accept, two borderline accept) and one borderline reject (AVhL).  During discussion, the reviewer AVhL think the paper can also be bumped up to borderline accept.

The AC agrees with the reviewers in the interesting nature of the work and recommends that the paper be accepted.  It will make for a nice contribution to ICLR.

**Additional Comments On Reviewer Discussion:**

no further comments

---

### Decision · Program_Chairs · 2025-01-22

Accept (Poster)